# Effects of *Alnus japonica* Hot Water Extract and Oregonin on Muscle Loss and Muscle Atrophy in C2C12 Murine Skeletal Muscle Cells

**DOI:** 10.3390/ph17121661

**Published:** 2024-12-10

**Authors:** Da Hyeon An, Chan Ho Lee, Yeeun Kwon, Tae Hee Kim, Eun Ji Kim, Jae In Jung, Sangil Min, Eun Ju Cheong, Sohyun Kim, Hee Kyu Kim, Sun Eun Choi

**Affiliations:** 1Department of Forest Biomaterials Engineering, Kangwon National University, Chuncheon 24341, Gangwon State, Republic of Korea; 202416560@kangwon.ac.kr (D.H.A.); lgh4107@gmail.com (C.H.L.); 2Dr.Oregonin Inc., #802 Bodeum Hall, Kangwondaehakgil 1, Chuncheon 24341, Gangwon State, Republic of Korea; kye0519@naver.com (Y.K.); kth02120@naver.com (T.H.K.); 3Industry Coupled Cooperation Center for Bio Healthcare Materials, Hallym University, Chuncheon 24252, Gangwon State, Republic of Korea; myej4@hallym.ac.kr (E.J.K.); jungahoo@hallym.ac.kr (J.I.J.); 4Division of Transplantation and Vascular Surgery, Department of Surgery, Seoul National University Hospital, Seoul 03080, Republic of Korea; surgeonmsi@gmail.com; 5College of Forest and Environmental Science, Kangwon National University, Chuncheon 24341, Gangwon State, Republic of Korea; ejcheong@kangwon.ac.kr (E.J.C.); rlathgus0613@naver.com (S.K.); 6Gangwon State Forest Science Institute, 24, Hwamokwon-gil, Chuncheon 24207, Gangwon State, Republic of Korea; dearkyu@korea.kr

**Keywords:** *Alnus japonica*, oregonin, hot water extract, apoptosis, muscle atrophy, sarcopenia

## Abstract

**Background/Objectives:** Sarcopenia is characterized by the loss of muscle mass and function, increases in mortality rate, and risk of comorbidities in the elderly. This study evaluated the effects of *Alnus japonica* hot water extract (AJHW) and its active compound, oregonin, on muscle atrophy and apoptosis in vitro. **Methods:** AJHW underwent phytochemical analysis. C2C12 cells were subjected to H_2_O_2_ and dexamethasone to induce oxidative stress and muscle loss, after which AJHW and oregonin were administered to assess their impacts on cell viability, apoptosis, muscle protein synthesis stimulation, and muscle protein degradation inhibition. Cell viability was assessed via an MTT assay, and apoptosis was analyzed by measuring Bcl-2, Bax, cleaved caspase-3, and cleaved PARP through Western blotting. Western blotting and RT-PCR were utilized to analyze MyoD, Myogenin, Atrogin-1, and MuRF1 protein and gene expression in a muscle atrophy model, as well as the Akt/mTOR and FoxO3α pathways. **Results:** AJHW was confirmed to contain oregonin, an active compound. AJHW and oregonin significantly increased cell viability and reduced apoptosis by upregulating Bcl-2 and downregulating Bax, cleaved caspase-3, and cleaved PARP. They significantly enhanced muscle protein synthesis through the upregulation of MyoD and Myogenin, while diminishing muscle degradation by downregulating Atrogin-1 and MuRF1. The activation of the Akt/mTOR pathway and inhibition of the FoxO3α pathway were also observed. **Conclusions:** AJHW and oregonin effectively prevented muscle cell apoptosis, promoted muscle protein synthesis, and inhibited muscle protein degradation in vitro. These results suggest that AJHW and oregonin could serve as therapeutic agents to prevent and treat sarcopenia.

## 1. Introduction

Muscle is categorized into skeletal, visceral, and cardiac types, with skeletal muscle responsible for crucial functions such as joint movement, posture maintenance, and heat production [1]. Muscle mass and bone density decline after approximately 30 years of age, with muscle loss occurring at a rate of 1–2% per year for individuals over 50 years and 3% per year for those over 60 years [2]. This loss contributes to sarcopenia, characterized by a reduction in both the number and size of muscle fibers. The term ‘sarcopenia’ denotes age-related loss of muscle mass and function, first identified by Irwin Rosenberg in 1989 [3]. Current research indicates that sarcopenia results from a multitude of factors, not just aging. These include inadequate exercise, hormonal imbalances, increased free radicals and inflammatory cytokines, mitochondrial damage, and obesity [4,5,6]. Sarcopenic obesity, where sarcopenia and obesity coexist, leads to greater reductions in muscle strength and function due to increased fat infiltration within the muscle. Additionally, the chronic inflammation associated with obesity exacerbates sarcopenia [7,8]. Beyond observable symptoms, such as diminished physical function and an elevated risk of falls and fractures, sarcopenia is associated with several severe diseases, including kidney, cardiovascular, liver, and breast cancers [9,10]. Consequently, in 2016, the World Health Organization (WHO) classified sarcopenia as an official ICD disease [11]. As the global elderly population grows, the prevalence of sarcopenia is also increasing. According to a survey on the prevalence of sarcopenia from the Korea National Health and Nutrition Examination Survey, the prevalence rates were found to be 19.2% in the 20–39-year age group, 29.1% in the 40–64-year age group, and 42.3% in those aged 65 years and older. This suggests that not only the elderly but also younger populations are at a potential risk [12].

The muscle atrophy and loss observed in sarcopenia are caused by excessive proteolysis, where muscle proteins are synthesized and degraded under the influence of various factors such as IGF-1, Akt, mTOR, MyoD, Myogenin, MuRF1, Atrogin-1, FoxO, etc. TNF-α and proinflammatory cytokines such as IL-6 and IL-1 activate the ubiquitin–proteasome pathway, promoting muscle protein degradation and leading to muscle atrophy [13,14,15,16]. Additionally, muscle protein synthesis is inhibited by the IGF-1/Akt/mTOR inhibitory mechanism of myostatin, which is expressed in muscle [17,18,19]. This promotes the activation of the FoxO transcription factor, increasing the expression of muscle-specific E3 ubiquitin ligases Atrogin-1 and MuRF1, which in turn promote muscle protein degradation, exacerbating sarcopenia [20,21]. Furthermore, oxidative stress induced by reactive oxygen species (ROS) triggers an inflammatory response through the NF-κB signaling pathway, increasing the expression of Atrogin-1 and MuRF1 and thereby promoting muscle protein degradation [22,23,24]. On the other hand, IGF-1 signaling activates Akt, resulting in the formation of phospho-Akt, which inhibits the activity of FoxO3α while decreasing the expression of Atrogin-1 and MuRF1, and thus promoting muscle protein synthesis and inhibiting degradation [2,25,26,27]. Thus, sarcopenia develops through complex mechanisms, and despite extensive research, no effective therapeutic agents have yet been identified to prevent or mitigate this condition [28]. As such, this study aimed to develop therapeutic agents for the prevention and mitigation of sarcopenia. 

On the other hand, various studies have recognized the safety, reliability, and validity of natural products for medicinal purposes [29]. *Alnus* species, widely distributed in the Northern Hemisphere, including Korea, the United States, Japan, and China, have been utilized in the treatment of enteritis, diarrhea, traumatic hemorrhage, fever, and inflammatory diseases, owing to their antipyretic, hemostatic, and astringent properties in traditional oriental medicine in Korea and China [30]; they are also used to treat various diseases in Western medicine [31].

Plants in the genus *Alnus* contain secondary metabolites with a variety of biological activity properties, including triterpenoids, flavonoids, polyphenols, steroids, and diarylheptanoids [32]. Notably, oregonin, hirsutanonol, and hirsutenone from the diarylheptanoid family exhibit diverse biological activities, such as antioxidant, anti-inflammatory, anticancer, anti-atopic, and hepatoprotective effects [33,34]. Previous studies have demonstrated that these compounds inhibit COX-2 and MMP-2 expression [35,36], suppress NO synthesis [37], and reduce NF-κB activity [38]. Furthermore, these compounds have shown anticancer activity by enhancing cytotoxicity in murine B16 melanoma cells and human SNU-C1 gastric cancer cells [39], anti-allergy effects through the inhibition of β-hexosaminidase secretion [40], and hepatoprotective activity against t-BHP-induced HepG2 cell injury [41].

Among the diarylheptanoid compounds, oregonin is found in high concentrations in *Alnus* species and serves as both a representative active substance and an indicator in the phytochemical analysis of *Alnus* species. Several biological activity studies using oregonin as a standard have been conducted [42,43,44]. Previous research has shown that oregonin inhibits nitric oxide and prostaglandin E2 synthesis [45], reduces the levels of inflammatory cytokines TNF-α, IL-1β, IL-2, IL-6, IL-10, and IL-12p70 [46], suppresses COX-2 and NF-κB activity [47], demonstrates anticancer activity in macrophages [48], and enhances NK cytotoxicity in B16-F10 melanoma cells [49]. Additionally, a recent study demonstrated that oregonin effectively scavenges reactive oxygen species (ROS) and inhibits and regulates oxidative stress-induced apoptosis in human breast papillary cells [50]. Consequently, we hypothesized that *Alnus japonica* extract and its main component, oregonin, could promote muscle protein synthesis and inhibit its degradation in myofibroblasts through antioxidant activity and by inhibiting myofibroblast apoptosis. Furthermore, a previous study indicated that *Alnus japonica* 50% ethanol extract and its major component, oregonin, alleviate sarcopenia by promoting muscle protein synthesis and impeding its degradation [51]. The previous study investigated various markers related to muscle synthesis and atrophy, including myosin heavy chain (MHC), MyoD, Myogenin, Myostatin, MuRF1, p38 MAPK, phospho-p38 MAPK, mTOR, phospho-mTOR, FoxO3a, phospho-FoxO3a, NF-κB, phospho-NF-κB, and MAFbX (Muscle Atrophy F-box, Atrogin-1). However, it did not thoroughly explore the expression of muscle-synthesis-related signaling markers, such as Akt and p-Akt, nor did it investigate apoptosis-related biomarkers, such as Bax, Bcl-2, PARP, and Caspase-3. Hence, we concluded that further research into the apoptotic mechanisms associated with sarcopenia is necessary. Furthermore, previous studies have utilized ethanol as an extraction solvent, which presents limitations for drug development. Organic solvents such as ethanol have faced criticism for their poor human safety profile, instability of active compounds, and limited commercial value [52,53]. Specifically, residual solvents, volatile organic compounds used or produced during drug manufacturing, have proven difficult to completely eliminate from production processes, posing significant challenges due to stringent regulatory requirements [54]. Consequently, we explored hot water extraction as an alternative to ethanol. Hot water extraction, a highly efficient method for extracting biological substances from natural products, is recognized for its stability and effectiveness. This technique, which employs high-temperature water to extract various polymer compounds, is notable for its low toxicity and environmental friendliness [55]. Additionally, hot water extraction has a long history of use in traditional oriental and folk medicine, where its safety and efficacy have been well documented [56]. 

Based on this background, this study aimed to evaluate the suitability of the hot water extraction method by comparing its efficiency and active compound content with those of the ethanol extracts of *Alnus japonica*. For this comparison, in this study, *Alnus japonica* was extracted using 50% ethanol (7 h) following the method used in previous research [51]. Additionally, *Alnus japonica* hot water extract was prepared using the hot water extraction process developed by our research team at a pilot scale (100 ± 5 °C, 4 h). The hot water extraction is considered more suitable than ethanol extraction, not only for human safety and compatibility, but also for the stability of active ingredients and economic feasibility. Moreover, hot water extraction offers substantial environmental and sustainability benefits. With the growing importance of Environmental, Social, and Governance (ESG) management in global industries, the demand for eco-friendly and sustainable production processes is increasing [57]. Hot water extraction, not relying on organic solvents, is well aligned with these demands. Furthermore, *Alnus japonica* hot water extract, as a pharmaceutical raw material, can significantly enhance competitiveness for global market entry [58]. 

In this paper, we initially conducted a phytochemical analysis of *Alnus japonica* ethanol extract (AJE) from existing studies, and *Alnus japonica* hot water extract (AJHW) by our research team, as previously mentioned, to identify and elucidate the differences. Subsequently, the researchers aimed to confirm the efficacy of the pilot-scale *Alnus japonica* hot water extract in enhancing muscle protein synthesis, reducing its degradation, and preventing muscle atrophy. To this end, we employed a dexamethasone-treated C2C12 skeletal muscle cell model to conduct detailed analyses of the protein and mRNA expression of pivotal muscle-related biomarkers including MuRF1, Myogenin, MyoD, FoxO, and mTOR. Furthermore, we intended to investigate further the action mechanism of *Alnus japonica* hot water extract on Akt, phospho-Akt, and Atrogin-1, critical biomarkers of sarcopenia previously underexplored in research. By exposing C2C12 cells to H_2_O_2_ to trigger oxidative stress and apoptosis, we verified the efficacy of *Alnus japonica* hot water extract in combating apoptosis-induced muscle loss and atrophy. The research team performed cross-validation at both protein and mRNA levels using Western blot assays and PCR analysis, thereby ensuring the study’s objectivity and reliability through a thorough comparative analysis. Consequently, this study sought to explore the potential effects and mechanisms of *Alnus japonica* hot water extract on muscle loss and atrophy through preclinical studies in four key areas: inhibition of oxidative stress, inhibition of apoptosis, enhancement of muscle protein synthesis, and reduction in muscle protein degradation. Ultimately, this study aimed to demonstrate the potential of *Alnus japonica* hot water extract and oregonin as promising candidates for new drug development to prevent and treat sarcopenia-related conditions.

## 2. Results

### 2.1. Phytochemical Analysis

#### 2.1.1. Qualitative Analysis of *Alnus japonica* Hot Water Extract (AJHW) and *Alnus japonica* Ethanol Extract (AJE) (TLC)

Using thin-layer chromatography (TLC), we conducted a qualitative analysis of AJHW and AJE. By comparing their Rf values and color reactions to the oregonin standard sample, it was confirmed that AJHW matched the standard sample (Figure 1).

#### 2.1.2. Measurement of the Molecular Weight of AJHW (LC-MS/MS)

The samples were analyzed by LC-MS/MS. The negative mode spectrum confirmed a molecular weight of 478.49 g/mol (Figure 2), matching that of oregonin, a known indicator material for the *Alnus* species.

#### 2.1.3. Quantitative Chromatographic Analysis of AJHW and AJE (HPLC)

Using the least-squares method, a relationship between concentration and peak area was established (R^2^ value). From six concentrations of oregonin, a standard calibration curve and curve equation were derived, yielding Y = 8139.8x − 7014.6 (R^2^ = 0.9997) increments of oregonin (Figure 3). The calibration curve exhibited excellent linearity (correlation coefficient ≥ 0.9997). The retention time of oregonin was 15.47 ± 0.01 min (Figure 4), and the average content of the oregonin was calculated as 38.62 ± 0.3 μg/mL (*n* = 3) in AJE and 53.52 ± 0.21 μg/mL (*n* = 3) in AJHW using this formula (Figure 5). 

### 2.2. The Impact of Alnus japonica Hot Water Extract (AJHW) and Oregonin on the Viability of C2C12 Cells

#### 2.2.1. Measurement of Cell Viability Under Normal Conditions

To investigate the cytotoxic effect of *Alnus japonica* hot water extract (AJHW) on C2C12 myoblasts, various concentrations of AJHW (0–1000 μg/mL) were added to the cell culture medium and incubated for 48 h, followed by an MTT assay. Cell viability significantly decreased at concentrations above 100 μg/mL compared to the control group (0 μg/mL) (Figure 6A). To further determine the non-cytotoxic concentration range of AJHW, additional concentrations (0, 2.5, 5, 10, 20, and 25 μg/mL) were examined, and cell viability was assessed. Notably, a slight reduction in cell viability occurred at concentrations of 20 μg/mL and 25 μg/mL, but these differences were not statistically significant compared to the control (Figure 6B). Consequently, the AJHW concentrations selected for further experiments were 0, 2.5, 5, 10, and 20 μg/mL. 

Similarly, to evaluate the cytotoxicity of oregonin on C2C12 myoblasts, concentrations ranging from 0 to 100 μg/mL were applied, and the cells were incubated for 48 h followed by an MTT assay. A significant reduction in cell viability was observed at concentrations of 50 μg/mL and higher (Figure 7). Thus, the maximum oregonin concentration used in subsequent experiments was limited to 10 μg/mL.

#### 2.2.2. Impact of H_2_O_2_-Induced Myoblast Damage

Hydrogen peroxide (H_2_O_2_) is a potent oxidizing agent that induces oxidative stress in vitro. To investigate the protective effects of *Alnus japonica* hot water extract (AJHW) and oregonin on H_2_O_2_-induced muscle cell damage, C2C12 myoblasts were exposed to 100 μM H_2_O_2_ to induce oxidative stress, followed by treatment with various concentrations of AJHW and oregonin. After 48 h of incubation, cell viability was measured. Compared to the untreated control, the H_2_O_2_-treated group exhibited a significant decrease in cell viability (Figure 8). AJHW treatment (2.5, 5, 10, and 20 μg/mL) increased cell viability in a dose-dependent manner, with the highest concentration of 20 μg/mL showing a 14.5% increase in cell viability compared to the H_2_O_2_-treated group, reaching 68.1 ± 1.4% (Figure 8A). Similarly, oregonin treatment (0.5, 1, 5, and 10 μg/mL) significantly enhanced cell viability at concentrations of 5 μg/mL and 10 μg/mL, resulting in a 10.2% increase to 67.1 ± 1% (Figure 8B). 

#### 2.2.3. Impact of Dexamethasone-Induced Myotube Damage

Dexamethasone (DEX), a commonly used glucocorticoid, can induce skeletal muscle degradation when misused and is widely used in vitro to model muscle atrophy. To investigate the effects of AJHW and oregonin on DEX-induced muscle cell damage, C2C12 myotubes were differentiated into myotubes by culturing in a differentiation medium for four days, followed by treatment with 5 μM DEX to induce muscle atrophy. After co-treatment with AJHW and oregonin for 24 h, the myotube viability was measured. DEX-treated cells showed a significant decrease in viability compared to the untreated control. AJHW treatment (2.5, 5, 10, and 20 μg/mL) significantly increased cell viability at concentrations of 5, 10, and 20 μg/mL compared to that in the DEX-treated group (Figure 9A). Additionally, oregonin treatment (0.5, 1, 5, and 10 μg/mL) significantly increased cell viability at a concentration of 10 μg/mL (Figure 9B).

### 2.3. Effects of Alnus japonica Hot Water Extract (AJHW) and Oregonin on Muscle Apoptosis Biomarkers

#### 2.3.1. Effects on H_2_O_2_-Induced Apoptosis in Myoblasts

Oxidative stress, known to induce apoptosis by causing DNA damage, typically occurs through programmed apoptosis under exposure to oxidative stressors like H_2_O_2_. This study quantified fragmented DNA using the Cellular DNA Fragmentation ELISA kit to evaluate the effects of *Alnus japonica* hot water extract (AJHW) and oregonin on oxidative stress-induced apoptosis. Compared to the untreated control, the H_2_O_2_-treated group exhibited a significant increase in apoptosis. AJHW treatment markedly reduced H_2_O_2_-induced apoptosis; specifically, concentrations of 2.5, 5, 10, and 20 μg/mL resulted in reductions of 4.5%, 13.3%, 24.9%, and 34.5%, respectively (Figure 10A). Similarly, oregonin treatment significantly mitigated apoptosis, with a 42.9% reduction observed at 10 μg/mL starting from 5 μg/mL (Figure 10B). Thus, the protective effects of AJHW and oregonin against H_2_O_2_-induced myoblast cell damage were confirmed.

#### 2.3.2. Effects of *Alnus japonica* Hot Water Extract (AJHW) and Oregonin on Bax, Bcl-2, Cleaved Caspase-3, and Cleaved PARP Protein Expression

Cells exposed to H_2_O_2_ primarily undergo apoptosis accompanied by mitochondrial damage, characterized by morphological changes such as cell shrinkage and DNA condensation. The Bcl-2 family plays a pivotal role in regulating apoptosis, particularly through the intrinsic pathway that maintains mitochondrial function and controls mitochondria-induced apoptosis. The Bcl-2 family comprises pro-apoptotic factors, such as Bax, and anti-apoptotic factors, such as Bcl-2. When the balance between these factors is disrupted, mitochondrial dysfunction occurs, leading to the release of cytochrome c into the cytoplasm, which activates subphase proteins that trigger apoptosis [59]. 

This study examined the effects of *Alnus japonica* hot water extract (AJHW) and oregonin on Bax and Bcl-2 protein expression in C2C12 myoblasts subjected to oxidative stress. Bax expression significantly increased in H_2_O_2_-treated groups compared to the control, but was significantly reduced to 0.19 ± 0.04 by AJHW treatment at a concentration of 20 μg/mL (Figure 11A,B). Conversely, oregonin had no significant effect on Bax expression in both H_2_O_2_-treated and untreated groups (Figure 11C,D). Bcl-2 expression was markedly suppressed in the H_2_O_2_-treated group compared to the control group (Figure 11E,F). However, AJHW treatment at 10 and 20 μg/mL significantly restored Bcl-2 expression to 1.96 ± 0.17 and 1.81 ± 0.1, respectively. Similarly, oregonin at 5 and 10 μg/mL significantly increased Bcl-2 expression to 1.10 ± 0.01 and 1.17 ± 0.03, respectively (Figure 11G,H).

Caspase proteases are crucial in initiating and executing apoptosis, with caspase-9 functioning as the primary initiator of the intrinsic apoptosis pathway. Activated caspase-9 initiates effector caspases, such as caspase-3 and caspase-7, which subsequently cleave target proteins like PARP, leading to DNA fragmentation and the completion of the apoptotic process. PARP, essential for DNA repair and genomic stability in healthy cells, is cleaved by caspase-3 during apoptosis, losing its repair ability. This cleavage is evident in cells undergoing apoptosis along with DNA fragmentation [60]. This study investigated the effects of *Alnus japonica* hot water extract (AJHW) on the expression of cleaved caspase-3 and cleaved PARP in H_2_O_2_-induced oxidative stress in C2C12 myoblasts. Compared to the untreated control group, the expression of cleaved caspase-3 significantly increased in the H_2_O_2_-treated group. AJHW treatment at 10 and 20 μg/mL significantly reduced this increased expression of cleaved caspase-3, decreasing levels to 0.6 ± 0.07 and 0.5 ± 0.04, respectively (Figure 12A,B). Similarly, oregonin treatment at concentrations of 5 and 10 μg/mL significantly reduced cleaved caspase-3 expression to 0.93 ± 0.01 and 0.6 ± 0.03, respectively (Figure 12C,D). The expression of cleaved PARP also increased significantly in the H_2_O_2_-treated group compared to the untreated control. However, AJHW treatment at concentrations of 5, 10, and 20 μg/mL significantly reduced cleaved PARP levels to 0.84 ± 0.03, 0.64 ± 0.07, and 0.5 ± 0.04, respectively (Figure 12E,F). Additionally, oregonin treatment at all tested concentrations (5, 10 μg/mL) also resulted in a significant reduction in cleaved PARP expression, to levels of 0.80 ± 0.02, 0.82 ± 0.02, and 0.79 ± 0.03, respectively (Figure 12G,H). This shows that AJHW and oregonin are effective in inhibiting muscle apoptosis.

### 2.4. Effects of Alnus japonica Hot Water Extract (AJHW) and Oregonin on Muscle-Synthesis- and Muscle-Degradation-Related Proteins and Gene Expression

#### 2.4.1. Effect of *Alnus japonica* Hot Water Extract (AJHW) and Oregonin on Dexamethasone-Induced Myotube Atrophy

To investigate the effects of *Alnus japonica* hot water extract (AJHW) and oregonin on dexamethasone-induced myotube atrophy, the diameters of C2C12 myotubes were measured after treatment. C2C12 cells were cultured in differentiation media for four days to form myotubes. Subsequently, 5 μM dexamethasone was introduced to induce muscle atrophy, followed by a 24 h treatment with AJHW and oregonin. After staining the myotubes with a MYH antibody for fluorescence imaging, their diameters were quantified.

Dexamethasone treatment significantly reduced the myotube diameters compared to the untreated control group. However, AJHW treatment substantially restored myotube diameters to levels similar to those in the untreated control group (Figure 13A). The diameter of the dexamethasone-treated group was reduced to 0.22 ± 0.01 μm compared to 0.51 ± 0.02 μm in the untreated group, and AJHW treatment increased the diameter 1.8-fold relative to the dexamethasone group (Figure 13C). Similarly, oregonin treatment markedly increased myotube diameter compared to the dexamethasone group, reaching levels similar to the untreated control group (Figure 13B). The diameter of the dexamethasone-treated group decreased to 0.12 ± 0.01 μm, while the untreated group measured 0.36 ± 0.02 μm. Oregonin treatment augmented the diameter 3.25-fold relative to the dexamethasone group (Figure 13D). It was notably observed that treatment with 10 μg/mL of oregonin increased muscle fiber diameter beyond that of the control group, thus confirming the significant protective effects of AJHW and oregonin against dexamethasone-induced myotube atrophy.

#### 2.4.2. Expression of Atrogin-1, MuRF1, Myogenin, and MyoD Protein in Dexamethasone-Treated C2C12 Myotubes

The primary mechanism for muscle atrophy involves the ubiquitin–proteasome signaling pathway, where muscle proteins are bound to ubiquitin, E2, E3 ligase, and proteasomes, facilitating the progression of muscle atrophy. Notably, MAFbx (Atrogin-1) and MuRF1, crucial E3 ubiquitin ligases, significantly accelerate muscle atrophy. Furthermore, myostatin suppresses Akt/mTOR/p70S6K signaling, inhibiting muscle growth. Conversely, myogenic regulators contribute to muscle regeneration and differentiation, with MyoD specifying the myoblasts—muscle-forming progenitors—and Myogenin-promoting muscle differentiation [61].

This research analyzed the impact of *Alnus japonica* hot water extract (AJHW) and oregonin on dexamethasone-induced muscle atrophy by examining the expression of proteins related to muscle degradation (Atrogin-1, MuRF1) and muscle synthesis (MyoD, Myogenin). The protein levels of Atrogin-1 and MuRF1 were notably higher in the dexamethasone-treated group compared to the control (Figure 14). However, AJHW at concentrations of 2.5, 5, 10, and 20 μg/mL markedly reduced Atrogin-1 expression (Figure 14A,B), and oregonin at concentrations of 1, 5, and 10 μg/mL significantly diminished its expression (Figure 14C,D). Furthermore, MuRF1 expression, increased by dexamethasone, was decreased by AJHW at concentrations of 5, 10, and 20 μg/mL (Figure 14E,F) and by oregonin at 1, 5, and 10 μg/mL (Figure 14G,H). 

Regarding muscle synthesis proteins, MyoD and Myogenin levels decreased in the dexamethasone-treated group compared to the control (Figure 15). AJHW treatment significantly increased MyoD expression at 5 and 20 μg/mL (Figure 15A,B), and oregonin elevated MyoD expression at 5 and 10 μg/mL (Figure 15C,D). Similarly, AJHW significantly boosted Myogenin levels at concentrations of 2.5, 5, and 10 μg/mL (Figure 15E,F), while oregonin raised Myogenin expression at 1 and 5 μg/mL (Figure 15G,H). Thus, it was confirmed that AJHW and oregonin reduced Atrogin-1 and MuRF1 expression while enhancing that of Myogenin and MyoD.

#### 2.4.3. Effects on Muscle-Degradation- and Synthesis-Related Gene Expression

The effects of *Alnus japonica* hot water extract (AJHW) and oregonin on the mRNA levels of muscle degradation genes (*Atrogin-1*, *MuRF1*) and muscle synthesis genes (*Myogenin*, *MyoD*) in dexamethasone-induced muscle atrophy were investigated. Compared to the control, dexamethasone-treated groups exhibited significantly increased mRNA expression of the muscle degradation genes *Atrogin-1* and *MuRF1*, while the mRNA expression of the muscle synthesis genes *Myogenin* and *MyoD* was significantly decreased. AJHW significantly mitigated the dexamethasone-induced increase in *Atrogin-1* mRNA expression at concentrations of 10 and 20 μg/mL, and similarly reduced the increase in *MuRF1* mRNA expression at 10 and 20 μg/mL. Furthermore, AJHW significantly reversed the dexamethasone-induced reduction in *MyoD* mRNA expression at 20 μg/mL, and the reduction in *Myogenin* mRNA expression was significantly reversed at AJHW concentrations of 5, 10, and 20 μg/mL (Table 1).

Treatment with oregonin notably reduced the dexamethasone-induced increase in *Atrogin-1* mRNA expression across all tested concentrations of oregonin. Similarly, oregonin significantly lowered the increase in *MuRF1* mRNA expression at a concentration of 10 μg/mL. Moreover, oregonin treatment at 5 and 10 μg/mL significantly raised the previously reduced *MyoD* mRNA expression, and elevated the reduction in *Myogenin* mRNA expression across all tested oregonin concentrations (Table 2). These findings demonstrate that AJHW and oregonin effectively inhibit muscle degradation and enhance muscle synthesis by decreasing the expression of *Atrogin-1* and *MuRF1* mRNA, and increasing the expression of *MyoD* and *Myogenin* mRNA.

#### 2.4.4. Akt/mTOR/FoxO-Signaling-Pathway-Related Muscle Protein Expression Investigation

Muscle protein synthesis is promoted through the regulation of mTOR by the IGF-1/PI3K/Akt signaling pathway, while muscle protein degradation is governed by the pathway involving forkhead box O (FoxO) and the ubiquitin–proteasome system. When FoxO is phosphorylated by Akt, it becomes inactive. However, when Akt activation decreases, FoxO becomes active in the nucleus, leading to an increased expression of muscle atrophy-related genes (Atrogin-1, MuRF1) and muscle protein degradation [62]. In this study, Western blot analysis was conducted to investigate the effects of AJHW and oregonin on Akt/FoxO signaling in dexamethasone-induced muscle atrophy. Compared to the control group, the expression of p-Akt was significantly reduced in the dexamethasone (DEX) group. However, AJHW at 5 and 10 μg/mL and all concentrations of oregonin significantly increased p-Akt expression. There was no significant difference in Akt expression between the control group and the DEX treatment group nor at any treatment concentration of AJHW and oregonin compared to the DEX treatment group. The p-Akt/Akt ratio, which reflects Akt activity, was significantly lower in the DEX group than in the control group. AJHW at concentrations of 5, 10, and 20 μg/mL (Figure 16A,B), and oregonin at 5 and 10 μg/mL significantly increased Akt activity (Figure 16C,D). Additionally, p-mTOR expression was significantly decreased in the DEX group compared to the control, but AJHW at 10 and 20 μg/mL and oregonin at 5 and 10 μg/mL significantly increased p-mTOR expression. There were no significant differences in mTOR expression between the control and DEX groups, nor between groups treated with AJHW at all concentrations. However, oregonin at 1 and 10 μg/mL showed a significant decrease in mTOR expression. The p-mTOR/mTOR ratio, which reflects mTOR activity, was significantly reduced in the DEX group compared to the control, but AJHW at 2.5, 10, and 20 μg/mL (Figure 16E,F) and oregonin at all concentrations significantly increased mTOR activity (Figure 16G,H). These results confirmed that AJHW and oregonin effectively promote protein synthesis by enhancing p-Akt and p-mTOR protein expression in muscle cells.

In the DEX-treated group, p-FoxO3α levels were significantly lower compared to the untreated control group, and increased significantly at AJHW concentrations of 10 and 20 μg/mL, and oregonin concentrations of 5 and 10 μg/mL. Conversely, FoxO3α levels were significantly higher in the DEX-treated group compared to the untreated control group, and decreased significantly at AJHW concentrations of 10 and 20 μg/mL and oregonin concentrations of 5 and 10 μg/mL. The p-FoxO3α/FoxO3α ratio decreased significantly in the DEX-treated group but increased across all AJHW and oregonin concentrations (Figure 17). These findings suggest that AJHW and oregonin can inhibit proteolysis by modulating FoxO3α levels in muscle proteins.

## 3. Discussion

Skeletal muscle performs critical functions such as movement, posture maintenance, and heat production. However, it diminishes and atrophies with age, leading to a decline in physical function due to sarcopenia. Consequently, this results in an increased risk of falls and fractures, as well as conditions including kidney disease, cardiovascular disease, and sarcopenic obesity [63]. Current treatments for sarcopenia encompass non-drug therapies, such as exercise and dietary modifications, as well as pharmacological interventions; however, no drugs are FDA-approved for sarcopenia, and some medications can cause side effects [64]. Therefore, there is a need to develop therapeutic agents that can alleviate sarcopenia with minimal side effects. On the other hand, functional materials derived from arboreal natural products have been traditionally used in treating various diseases due to their active compounds with diverse biological activities and minimal side effects. Among these, *Alnus japonica* has been utilized to treat antipyretic, hemostatic, astringent, and inflammatory diseases in traditional Korean medicine [30]. Specifically, previous studies have revealed that oregonin, an active compound from *Alnus japonica*, offers potent physiological activities, including anti-cancer, anti-inflammatory, anti-atopy, antioxidant, and anti-apoptotic effects [40,42,43,44,46,49,50]. Additionally, a prior study reported that the ethanol extract of *Alnus japonica* inhibited the proximal axis in dexamethasone-induced C2C12 myotubes [51]. Nonetheless, we identified various studies indicating that ethanol extraction may be inadequate in several aspects such as cost, human safety, and environmental sustainability for future drug development [54,65,66,67,68], prompting us to explore hot water extraction as an alternative method. 

Before assessing the impact on muscle atrophy, the researchers performed a phytochemical analysis using TLC and LC-MS/MS to confirm the presence of oregonin in AJHW from alder hot water extract. To optimize this process, extraction conditions at the pilot scale were established, revealing that a 4 h extraction yielded the most stable content levels. Ethanol extraction, following a previous study’s method, was conducted over 7 h, with dextrin subsequently mixed and spray-dried. HPLC analysis of the extracted products demonstrated that the oregonin content was 38.6% higher in AJHW compared to AJE. Furthermore, in the case of hot water extraction, the extraction time was reduced to 4 h, which was 3 h shorter than the ethanol extraction time. This reduction is a significant discovery, as it can greatly enhance production efficiency and cost-effectiveness, making it an important consideration for future drug development. Building on this, our research team focused on identifying the effects and mechanisms of muscle loss and atrophy using AJHW. By conducting in vitro studies, we aimed to confirm its potential as a new drug ingredient across four key categories: oxidative stress inhibition, apoptosis inhibition, promotion of muscle protein synthesis, and muscle protein degradation inhibition.

To assess their effectiveness in mitigating sarcopenia, we investigated their protective effects against oxidative stress from H_2_O_2_- and dexamethasone-induced myocyte atrophy. AJHW (2.5, 5, 10, and 20 μg/mL) and oregonin (0.5, 1, 5, and 10 μg/mL) were found to increase the cell viability of myocytes, which had been decreased by H_2_O_2_ treatment by approximately 14.5% and 10.2%, respectively, at their highest concentrations. Additionally, cell viability improvements were observed at about 20.2% and 9.1%, respectively, at their maximum concentrations against myocyte damage caused by dexamethasone. These results indicate that AJHW and oregonin are effective in restoring myocyte cell viability, thereby mitigating damage to skeletal muscle cells.

H_2_O_2_ (hydrogen peroxide) is a potent oxidant that induces oxidative stress in vitro, leading to apoptosis. This apoptotic pathway is regulated by Bcl-2 family proteins, with the balance between pro-apoptotic factors, such as Bax, and anti-apoptotic factors, such as Bcl-2, playing a crucial role in determining the outcome of apoptosis [69]. Upon exposure to oxidative stress such as H_2_O_2_, Bax modifies mitochondrial permeability, releases cytochrome c, activates caspase-9 and caspase-3, and triggers apoptosis [70]. During this process, cleaved caspase-3 and cleaved PARP serve as principal apoptotic mechanisms in cell damage induced by oxidative stress [71]. In this experiment, we assessed apoptosis in C2C12 myoblasts and observed that AJHW and oregonin significantly reduced apoptosis in C2C12 myoblasts, which was elevated following H_2_O_2_ treatment. A recent study demonstrated that *Ulmus macrocarpa* extract and its active compound, catechin 7-O-β-D-apiofuranoside, inhibited apoptosis by 18.5% and 8.3% at concentrations of 200 μg/mL and 100 μg/mL, respectively. In contrast, AJHW and oregonin exhibited a more pronounced reduction in apoptosis, at rates of 34.5% (AJHW 20 μg/mL) and 42.9% (ORE 10 μg/mL) at their highest treatment concentrations of 20 μg/mL and 10 μg/mL, indicating that AJHW and oregonin are highly effective in inhibiting apoptosis at lower concentrations compared to other natural products [72]. 

Furthermore, the expression of apoptosis-related proteins in a cell model of H_2_O_2_-induced myocyte injury was examined. The expression of Bax, a pro-apoptotic protein, in myoblasts increased by 42% after H_2_O_2_ treatment. This increase was significantly reduced by AJHW treatment, with a decrease of about 81% at the highest AJHW concentration (20 μg/mL). In contrast, oregonin treatment did not significantly induce Bax expression in H_2_O_2_-treated C2C12 myoblasts. The expression of Bcl-2, an apoptosis inhibitor, was decreased by approximately 43% and 67% after H_2_O_2_ treatment and was significantly increased by AJHW and oregonin treatments, showing increases of approximately 96% and 17% at 10 μg/mL for AJHW and oregonin, respectively. We also investigated the expression of cleaved caspase-3 and cleaved PARP, which are pro-apoptotic signaling proteins. The results indicated that the expression of cleaved caspase-3 was significantly increased by H_2_O_2_ treatment and significantly decreased by AJHW and oregonin treatments. The expression of cleaved caspase-3 was decreased by 40%, both at 10 μg/mL of AJHW and the highest oregonin concentration (ORE 10 μg/mL), aligning with the control levels. The expression of cleaved PARP was notably increased by H_2_O_2_ treatment and significantly reduced by AJHW and oregonin treatments, with decreases of 36% (AJHW 20 μg/mL) and 21% (ORE 10 μg/mL), respectively. These findings suggest that AJHW mitigates H_2_O_2_-induced myocyte damage and alleviates myocyte injury by bolstering the expression of Bcl-2, a critical regulator of myocyte damage, and reducing the expression of Bax, cleaved caspase-3, and cleaved PARP. Meanwhile, it was confirmed that only the correlation of other apoptosis-related biomarkers, except Bax, was observed with oregonin. This suggests that oregonin does not influence Bax expression in these cells. According to previous studies, the efficacy of natural product extracts may be due to the synergistic effects of various compounds in the extract [73]. In this study, the results related to Bax expression were also presumed to be due to the synergistic effect of trace compounds other than oregonin contained in AJHW.

Dexamethasone is commonly used to induce proximal shrinkage in C2C12 myotubes, which results in an increased expression of several genes involved in the ubiquitin–proteasome pathway [74]. Dexamethasone-induced muscle contraction leads to a decrease in myofiber number and diameter, thereby reducing overall muscle mass [75]. In this study, we employed a C2C12 myotubes model of dexamethasone-induced proximal contraction to explore the mechanisms through which AJHW and oregonin counteract muscle loss and atrophy. AJHW and oregonin significantly increased the diameter of myotubes atrophied by dexamethasone treatment, with oregonin treatment at 10 μg/mL restoring myotube diameter to levels comparable to the control group.

Subsequently, the expression of muscle-synthesis- and -degradation-related proteins such as Atrogin-1, MuRF1, MyoD, Myogenin, Akt, p-Akt, mTOR, p-mTOR, FoxO3α, and p-FoxO3α was analyzed and cross-validated at both mRNA and protein levels using RT-PCR and Western blot analyses. Many studies have investigated the mechanisms of muscle protein degradation and synthesis, revealing that preventing and ameliorating muscle atrophy involves not only inhibiting muscle degradation, but also enhancing muscle synthesis [24]. Satellite cells are crucial for muscle regeneration, and myogenic regulatory factors (MRFs) such as MyoD and Myogenin are essential in this process [76]. Additionally, Atrogin-1 and MuRF1 enhance muscle protein degradation, a primary contributor to muscle loss, making them significant biomarkers in sarcopenia treatment [77]. The protein and mRNA expression of myolysis-related factors Atrogin-1 and MuRF1 increased after dexamethasone treatment but significantly decreased following treatment with AJHW and oregonin. Notably, Atrogin-1 protein expression decreased by 31% (AJHW 20 μg/mL) and 20% (ORE 10 μg/mL) at the highest concentrations of AJHW and oregonin, respectively. Similarly, MuRF1 decreased by 24% (AJHW 20 μg/mL) and 30% (ORE 10 μg/mL) at the highest treatment concentrations of AJHW and oregonin, respectively, showing that AJHW treatment resulted in a protein expression comparable to the control group. The protein and mRNA expression of myogenesis-related factors MyoD and Myogenin were reduced by dexamethasone treatment, but this reduction was significantly reversed by AJHW and oregonin treatment. Specifically, MyoD expression increased by 15% and 30% at 5 μg/mL of AJHW and oregonin, respectively, and Myogenin expression increased by 25% and 29% at 2.5 μg/mL of AJHW and 1 μg/mL of oregonin, respectively, affirming that the expression of MyoD and Myogenin matched that of the control group. These results suggest that AJHW prevented DEX-induced atrophy in C2C12 myotubes by inhibiting muscle protein degradation (Atrogin-1, MuRF1) and promoting synthesis (MyoD, Myogenin). Meanwhile, the earlier study demonstrated that *Ulmus macrocarpa* extract (UME) significantly reduced Atrogin-1 expression at a treatment concentration of 100 μg/mL, decreased MuRF1 expression at 50 μg/mL, and increased the expression of Myogenin and MyoD at 50 μg/mL [72]. Additionally, another former study found that *Cibotium barometz* extract markedly suppressed MurF1 expression at 100 μg/mL [78], indicating that AJHW and oregonin exhibit significant activity at low doses in modulating proteins related to muscle degradation and synthesis.

mTOR is activated by Akt and promotes protein synthesis, while FOXO transcription factors mediate protein degradation [79]. Specifically, FoxO3α is a key regulator of muscle protein degradation. When phosphorylated by Akt, FoxO3α is excluded from the nucleus and repressed, and upon dephosphorylation, it translocates to the nucleus as a transcription factor, subsequently upregulating E3 ubiquitin ligases such as MuRF1 and Atrogin-1 [80]. Consequently, the Akt/Fox3a signaling pathway serves as a potential target for preventing muscle atrophy. In this study, we observed that dexamethasone treatment inhibited the activation of p-Akt and p-mTOR, whereas AJHW and oregonin treatment significantly enhanced them. Notably, treatment with 5 μg/mL of AJHW and 5 μg/mL of oregonin increased the p-Akt/Akt ratio by 33% and 31%, respectively, similar to the control group. Furthermore, the p-mTOR/mTOR ratio increased by 49% and 34% with 5 μg/mL of AJHW and 1 μg/mL of oregonin, respectively, matching the control group levels. Thus, AJHW and oregonin significantly inhibit muscle protein synthesis and atrophy at low doses. These findings suggest that AJHW at low doses is more effective compared to *Ulmus macrocarpa* extract (UME), which increased p-Akt expression significantly at 100 μg/mL and p-mTOR at 50 μg/mL [72], and *Cibotium barometz* extract, which significantly increased the expression of p-Akt and p-mTOR at 50 μg/mL [78]. Dexamethasone treatment increased the expression of FoxO3α, a factor related to muscle proteolysis, which was significantly decreased by AJHW and oregonin treatment. Moreover, the p-FoxO3α/FoxO3α ratio increased by 66% and 64%, respectively, at the highest treatment concentration of AJHW and oregonin, reaching levels seen in the control group. The increases in p-Akt/Akt and p-mTOR/mTOR ratios indicate that AJHW and oregonin treatments promote protein synthesis, while the increase in the p-FoxO3α/FoxO3α ratio suggests a reduction in FoxO3α expression, thereby inhibiting muscle protein degradation. These results indicate that both oregonin and AJHW exert their effects by modulating key intracellular signaling pathways that are critical to muscle atrophy regulation. In other words, the expression levels of p-Akt/Akt, p-mTOR/mTOR, and p-FoxO3α/FoxO3α in this study indicate that AJHW and oregonin enhance the anabolism and reduce the catabolism of muscle-specific proteins. However, this study has limitations in that it was unable to clearly elucidate the molecular mechanisms related to the intracellular effects of oregonin. Therefore, further studies are warranted to elucidate the precise molecular mechanisms of oregonin, particularly by identifying cytosolic or nuclear enzymes, or transcription factors specifically bound to or modified by oregonin. Furthermore, it has been confirmed that other diarylheptanoid compounds, in addition to oregonin, are present in *Alnus japonica* hot water extract, and further research into the mechanisms of action of these compounds is also needed.

In conclusion, this study confirmed that AJHW and oregonin enhance the expression of muscle-synthesis proteins while diminishing the expression of proteins associated with muscle atrophy, by inhibiting apoptosis and loss induced by H_2_O_2_ and dexamethasone. Notably, AJHW and oregonin exhibited significant effects at low concentrations, setting them apart from existing natural products that necessitate higher concentrations to be effective. These characteristics promise substantial cost efficiency and safety, positioning them as potential key natural products for developing treatments for sarcopenia, subject to further in vivo validation.

## 4. Materials and Methods

### 4.1. Plant Extract Materials

This study utilized an *Alnus japonica* (AJ) extract; oregonin was isolated and purified from it. *Alnus japonica* bark, used in this research, was procured from Seoul Yakryeong Market and verified by Professor Choi (Department of Forest Biomaterials Engineering, Kangwon National University). The bark was cleansed, rinsed to eliminate impurities, and then used as an experimental material. A specimen of the *Alnus japonica* hot water extract (AJHW-2023-09) is preserved at the Department of Forest Biomaterials Engineering, Kangwon National University.

### 4.2. Extraction Method

#### 4.2.1. Pilot-Scale Hot Water Extraction of the Bark of *Alnus japonica*

*Alnus japonica* hot water extract powder was produced on a pilot scale using the following process: 300 kg of raw material was extracted with 3000 L of distilled water at 100 ± 5 °C for 4 h before concentration. The concentrated procedure employed a natural circulation technique, and 136.55 kg of *Alnus japonica* concentrate was subsequently obtained by concentrating further at 60 ± 5 °C (yield 45.52%). Additionally, 10% dextrin was added to 300 mL of raw material, then freeze-dried to yield 8.56 g of *Alnus japonica* hot water extract powder (yield 81.52%, lot no. 230901, lot no. DJTF-11872). This extraction process was conducted by ChuncheonBio Co., LTD (Chuncheon, Republic of Korea) and DanjoungBio Co., LTD (Wonju, Republic of Korea).

#### 4.2.2. Preparation of Ethanol Extraction

The ethanol extract was prepared as per the method outlined in earlier studies to produce an ethanol extract of Alder tree bark. AJ bark weighing 500 g was extracted with 50% ethanol (5 L) for 7 h at 80 ± 5 °C. The AJ extract was then sterilized, evaporated under reduced pressure at 65 °C, blended with maltodextrin DE20 at a 1:1 (*w*/*w*) ratio, and spray-dried in a pilot-scale spray dryer to achieve a yield of 20% [51].

### 4.3. Phytocemical Analysis

#### 4.3.1. Qualitative Analysis of Oregonin (TLC)

Thin-layer chromatography was performed to identify AJHW. Standard samples and AJHW, each weighing 2.5 mg, were dissolved in 1 mL of methanol to prepare 2500 ppm solutions. Using a silica gel plate, both sample types and the standard sample were spotted, and the plate was developed using chloroform/methanol/water (CMW) at a 70:30:4 ratio. After the silica gel plate was completely dried, it was analyzed at 254 nm with a UV lamp, using three coloring reagents: 10% H_2_SO_4_, *ρ*-anisaldehyde H_2_SO_4_, and FeCl_3_.

#### 4.3.2. Separation and Purification of Oregonin

The purification and isolation of oregonin were conducted using the method described by Choi, S.E., 2013 [42] (Figure 18).

**Oregonin(1)** is a brown amorphous powder with a negative LC-MS/MS result: m/z 477 [M–H]^−^.

^1^H-NMR (700 MHz, MeOH-*d*_4_): 6.657 (H-2″, 1H, d, *J* = 2.8 Hz), 6.645 (H-5′, 1H, d, *J* = 2.8 Hz), 6.616 (H-5″, 1H, d, *J* = 2.1 Hz), 6.607 (H-2′, 1H, d, *J* = 2.1 Hz), 6.492 (H-6″, 1H, d, *J* = 2.1 Hz), 6.470 (H-6′, 1H, d, *J* = 2.1 Hz), 4.217 (xyl-1, 1H, d, *J* = 7.7 Hz), 3.865 (H-5, 1H, d, *J* = 5.6 Hz), 3.312 (xyl-1, 1H, d, *J* = 1.4 Hz), 3.310 (xyl-5e, 1H, d, *J* = 2.1 Hz), 3.303 (xyl-4, 1H, d, *J* = 4.2 Hz), 3.168, 3.124 (2H, m, xyl-3, 5a), 2.85–2.45 (8H in total, H-1,2,4,7), 1.778–1.702 (2H in total, m H-6) [40]. 

^13^C-NMR (150 MHz, MeOH-*d*_4_): 146.27 (C-3″), 146.18 (C-3′), 144.56 (C-4′), 144.29 (C-4″), 135.28 (C-1′), 133.17 (C-1″), 120.82 (C-6′), 120.72 (C-6″), 116.74 (C-5″), 116.66 (C-5′), 115.51 (C-2″), 115.40 (C-2′), 104.44 (Xyl-1), 78.03 (Xyl-3), 76.47 (C-5), 75.24 (Xyl-2), 71.40 (Xyl-4), 67.09 (Xyl-5), 28.66 (C-4), (C-2), (C-6), (C-7), 30.23 (C-1), (C-3) [40,81].

#### 4.3.3. Measurement of Oregonin Molecular Weight (LC-MS/MS)

LC-MS/MS analysis was conducted using AJHW and the AB SCIEX (QTRAP 4500, Billerica, MA, USA) model. The column conditions included a Hector C18 column (5 μm) paired with a Phenomenex KJ0-4282 guard column. Mobile phases comprised 0.9% acetic acid in water (A) and acetonitrile (B). Wavelength analysis was conducted at 280 nm, with a flow rate of 1 mL/min over 40 min. The gradient program included the following: 0–0 min, 10% B; 0–12 min, 25% B; 12–24 min, 40% B; 24–25 min, 10% B; 25–40 min, 10% B.

#### 4.3.4. Quantitative Chromatographic Analysis of Oregonin (HPLC)

The HPLC analysis of AJHW utilized a Waters 2695 system model and a 2487 dual-λ absorbance detector (Waters, Milford, MA, USA). The setup included a Hector C18 column (5 μm) and a Phenomenex KJ0-4282 guard column. A 20 μL sample was injected, the wavelength set at 280 nm, with the analysis performed at a flow rate of 1 mL/min for 40 min using 0.9% acetic acid in water (A) and acetonitrile (B) as the mobile phases. The gradient program was as follows: 0–0 min, 10% B; 0–12 min, 25% B; 12–24 min, 40% B; 24–25 min, 10% B; 25–40 min, 10% B. For quantitative analysis, oregonin, previously isolated from *Alnus japonica* extract, served as the standard. Oregonin (1 mg) was dissolved in 1 mL of methanol. This solution was then diluted to concentrations of 100, 50, 25, 10, 5, and 1 μg/mL.

### 4.4. Cell Culture and Treatment

Mouse skeletal-muscle-derived myoblasts, known as C2C12 cells, were acquired from the American Type Culture Collection (ATCC, Manassas, VA, USA). These cells were maintained in a moisture-controlled CO_2_ incubator (5% CO_2_/95% air) at 37 °C. The growth medium consisted of Dulbecco’s modified eagle medium (DMEM) enriched with 10% fetal bovine serum (FBS), 100 units/mL penicillin, and 100 μg/mL streptomycin. Upon reaching approximately 80% confluence, the culture vessel was rinsed with phosphate-buffered saline (PBS, pH 7.4). Subsequently, trypsin–2.65 mM EDTA was employed to detach and subculture the cells. The medium was replaced every two days. To induce myotube formation, C2C12 cells were cultured in myotube differentiation medium containing 2% horse serum (HS) in DMEM, with medium changes every two days.

### 4.5. Cell Viability Assay

#### 4.5.1. Measurement of C2C12 Myoblast Viability Using MTT Assay

The viability of C2C12 myoblasts was assessed using an MTT assay (Denizot F and Lang R, J Immunological Method 89: 271-277, 1986). C2C12 myoblasts were seeded at 5 × 10^4^ cells/well in a 24-well plate and cultured for 24 h. Subsequently, the cell culture medium was replaced with a medium containing varying concentrations of AJHW and ORE, and the cells were cultured for an additional 48 h. The culture medium was then replaced with 1 mg/mL MTT solution, and the cells were further cultured for 2 h. After eluting the formazan produced in the viable cells with isopropanol, absorbance was measured at 570 nm to assess cell viability. 

#### 4.5.2. Measurement of Protective Effects Against H_2_O_2_-Induced Myoblast Damage

C2C12 cells were seeded at 5 × 10^4^ cells/well in a 24-well plate and cultured for 24 h. After culturing C2C12 myoblasts for 24 h, 100 μM H_2_O_2_ was administered to induce myocyte damage, followed by the addition of test substances at various concentrations, along with 100 μM H_2_O_2_ to explore the protective effects of AJHW and oregonin against myocyte damage. After a further 48 h of culture, cell viability was assessed using an MTT assay, as previously described.

#### 4.5.3. Measurement of Protective Effects Against Dexamethasone-Induced Myotube Damage

C2C12 cells were seeded at 5 × 10^4^ cells/well in a 24-well plate and cultured for 24 h. The medium was then replaced with a myocyte differentiation medium to initiate the differentiation of C2C12 cells into myotubes over 4 days. Following this, 5 μM dexamethasone was added to induce myocyte atrophy, and various concentrations of test substances were concurrently administered with 5 μM dexamethasone to evaluate the protective effects of AJHW and oregonin on myotube damage. The cells were cultured for another 24 h, after which cell viability was determined using an MTT assay, following the same protocol as earlier.

### 4.6. Effects of Alnus japonica Hot Water Extract (AJHW) and Oregonin on Muscle Apoptosis Biomarkers

#### 4.6.1. Evaluation of Apoptosis in H_2_O_2_-Induced Myoblasts

To assess the effects of different test compounds on hydrogen peroxide-induced apoptosis in myoblasts, C2C12 cells were seeded in 24-well plates at a density of 5 × 10^4^ cells per well and cultured for 24 h. The myoblasts were then subjected to 100 μM H_2_O_2_ to induce cellular damage. To examine the protective effects against this injury, five different test substances were administered at varying concentrations along with 100 μM H_2_O_2_, followed by a further 48 h incubation. Apoptosis was quantified using the Cellular DNA Fragmentation ELISA kit from Sigma-Aldrich (St. Louis, MO, USA), which detects 5′-bromo-2′-deoxyuridine (BrdU)-labeled DNA, according to the manufacturer’s instructions.

#### 4.6.2. Western Blot Analysis of Apoptosis Biomarkers in C2C12 Myoblasts

C2C12 myoblasts were seeded in a 100 mm dish at 1 × 10^6^ cells/well and cultured for 24 h. After 24 h of culturing, the cells were exposed to varying concentrations of AJHW and oregonin along with 100 μM H_2_O_2_, and cultured for an additional 24 h. Cells were then harvested, and a total cell lysate was prepared using the method described above. The protein content in the lysate was quantified and analyzed by SDS-PAGE, followed by Western blotting with various antibodies (Bax, Bcl-2, cleaved caspase-3, cleaved PARP, and β-actin antibody (Cell Signaling Technology, Beverly, MA, USA)). Protein bands were detected using enhanced chemiluminescence with Luminata^TM^ Forte Western HRP Substrate (Millipore, Burlington, MA, USA). Protein expression levels were quantified with an ImageQuant^TM^ LAS 500 imaging system (GE Healthcare Bio-Sciences AB, Uppsala, Sweden).

### 4.7. Effects of Alnus japonica Hot Water Extract (AJHW) and Oregonin on Muscle-Synthesis- and Muscle-Degradation-Related Proteins and Gene Expression

#### 4.7.1. Measurement of Myotube Diameter

C2C12 cells were cultured on cover glasses in 24-well plates at 5 × 10^4^ cells/well for 24 h. To promote differentiation into myotubes, the medium was replaced with a myocyte differentiation medium, and differentiation proceeded for 4 days. Subsequently, 5 μM dexamethasone was administered to induce myocyte atrophy, and the cells were further treated with various concentrations of the test substance for 24 h to evaluate its effect on alleviating myocyte atrophy. After culturing, the medium was removed, cells were washed with PBS, fixed with 4% paraformaldehyde and 0.1% Triton X-100, blocked with 5% BSA/TBST, and then subjected to primary antibody treatment (MYH7, Santa Cruz, Dallas, TX, USA). Tissues were stained with a secondary antibody (Anti-mouse IgG-Alexa-594, ThermoFisher Scientific, Waltham, MA, USA) and counterstained with 4′,6-diamidino-2-phenylindole (DAPI, Sigma-Aldrich, St. Louis, MO, USA). Muscle atrophy was assessed using an optical microscope (Carl Zeiss, Jena, Germany), and the thickest portion of each myotube was measured for diameter using ImageJ software (National Institutes of Health, Bethesda, MD, USA, Version 1.54).

#### 4.7.2. Real-Time Reverse Transcription Polymerase Chain Reaction (RT-PCR) Analysis

C2C12 cells were seeded in 24-well plates at 5 × 10^4^ cells/well and cultured for 24 h. To induce differentiation into myotubes, the culture medium was switched to the differentiation medium, and the cells were then cultured for four days. Subsequently, the cells were treated with various concentrations of AJHW and oregonin along with 5 μM dexamethasone for 24 h. Afterwards, the cells were harvested, and total RNA was isolated using the RNeasy^®^ Plus Mini kit (QIAGEN, Valencia, CA, USA). The quality and purity of the total RNA were assessed using a micro-volume spectrophotometer (BioSpec-nano, SHIMADZU, Kyoto, Japan); RNA with an OD260/280 ratio of 1.8 or higher was considered suitable for further experiments.

cDNA was synthesized from total RNA (2 μg) using the HyperScript^TM^ RT master mix kit (GeneAll Biotechnology, Seoul, Republic of Korea). Real-time PCR was then conducted with the Rotor-Gene 300 PCR (Corbett Research, Mortlake, Australia) using the Rotor-Gene^TM^ SYBR Green kit (QIAGEN). Primer details for this experiment are provided in Table 3. Gene expression was quantitatively analyzed using the Rotor-Gene 6000 Series System Software program version 6 (Corbett Research).

#### 4.7.3. Western Blot Analysis of Muscle Protein Expression in C2C12 Myotubes

C2C12 myoblasts were plated in a 100 mm dish at a density of 1 × 10^6^ cells/well and incubated for 24 h. To cause differentiation into myotubes, the culture medium was replaced with the differentiation medium and the cells were cultured for four days. Following differentiation, the cells were treated with various concentrations of AJHW (*Alnus japonica* hot water extract) and oregonin, along with 5 μM dexamethasone, and incubated for 24 h. The cells were then collected and lysed using a lysis buffer (20 mmol/L Hepes, pH 7.5, 150 mmol/L NaCl, 1% Triton X-100, 1 mmol/L EDTA, 1 mmol/L EGTA, 100 mmol/L NaF, 10 mmol/L sodium pyrophosphate, 1 mmol/L Na3VO4, 20 μg/mL aprotinin, 10 μg/mL antipain, 10 μg/mL leupeptin, 80 μg/mL benzamidine HCl, 0.2 mmol/L PMSF). The cells were homogenized and the total cell lysate was obtained by centrifugation. The protein content of this lysate was measured using a BCA protein assay kit (Thermo Scientific). A 50 μg sample of the total cell lysate was subjected to 10% sodium dodecyl sulfate–polyacrylamide gel electrophoresis (SDS-PAGE) and transferred to a polyvinylidene difluoride (PVDF) membrane (Millipore). The membrane was blocked with 5% skim milk–TBST (20 mmol/L Tris-HCl, pH 7.5, 150 mmol/L NaCl, 0.1% Tween 20) for one hour, then incubated with primary antibodies (Atrogin-1, MuRF1 (Santa Cruz Biotechnology); MyoD, Myogenin (Abcam, Cambridge, MA, USA); phospho-Akt, Akt, phospho-FoxO3α, FoxO3α, phospho-mTOR, mTOR, β-actin (Cell Signaling Technology)) for 6 h or 1 h at room temperature with shaking. Subsequently, the membrane was incubated with horseradish peroxidase (HRP)-linked anti-rabbit IgG or HRP-linked anti-mouse IgG for one hour with shaking, and the protein bands were visualized using the enhanced chemiluminescence method with Luminata™ Forte Western HRP Substrate (Millipore). Protein expression levels were quantified using the ImageQuant™ LAS 500 imaging system (GE Healthcare Bio-Sciences AB). 

### 4.8. Statistical Analysis

All analysis values are expressed as the mean ± S.E.M. The collected results were analyzed using the GraphPad Prism 5.0 software (GraphPad Software, San Diego, CA, USA). Student’s *t*-test and a one-way analysis of variance (ANOVA) were used to compare differences between the control group and the test substance treatment group. It was considered statistically significant only when *p* < 0.05 or higher.

## 5. Conclusions

The incidence of sarcopenia, which is characterized by a decline in skeletal muscle mass and function, is increasing, particularly among the elderly. Consequently, there is a need to develop treatments for sarcopenia that have fewer side effects. Previous studies have shown that *Alnus japonica* possesses anti-inflammatory, antioxidant, anti-cancer, and anti-apoptotic properties, and its efficacy has been proven in traditional oriental medicine. In this study, we confirmed that *Alnus japonica* hot water extract (AJHW) and its active component, oregonin, effectively inhibited muscle cell apoptosis and muscle loss induced by H_2_O_2_ and dexamethasone. 

In apoptosis assessments, treatment with AJHW and oregonin significantly increased the expression of Bcl-2, while significantly reducing the expression of apoptosis-related markers, including Bax, caspase-3, and cleaved PARP. Additionally, in evaluating muscle protein synthesis and degradation inhibition, AJHW and oregonin treatment significantly upregulated the expression of MyoD and Myogenin, the genes involved in muscle formation, while significantly downregulating the expression of Atrogin-1 and MuRF1, the genes associated with muscle atrophy. Furthermore, muscle protein synthesis pathways, such as p-Akt/Akt and p-mTOR/mTOR, were activated, and the p-FoxO3α/FoxO3α ratio increased, indicating an inhibitory effect on muscle protein degradation. In particular, AJHW and oregonin demonstrated a remarkable inhibition of muscle loss even at low concentrations, distinguishing them from existing natural product extracts that depend on high concentrations. 

Phytochemical analysis confirmed that AJHW is an extract primarily composed of oregonin. Additionally, the analysis revealed that AJHW contained a higher oregonin content than the ethanol extracts, and the stability of the active components, oregonin, was confirmed. These findings suggest that *Alnus japonica* can be economically and environmentally extracted using the hot water extraction method, with the extracted components exhibiting high stability, making them suitable materials for pharmaceutical development. Moreover, cost-saving effects can be anticipated in future research and pilot-scale production.

In conclusion, this study confirmed that AJHW and oregonin not only promoted the expression of muscle synthesis proteins, but also reduced the expression of muscle-atrophy-related proteins by inhibiting muscle apoptosis, muscle loss, and muscle atrophy induced by H_2_O_2_ and dexamethasone. In particular, AJHW and oregonin demonstrated excellent muscle loss inhibition effects even at low concentrations, differing from existing high-concentration-dependent natural product-based treatments. These properties suggest significant potential for cost effectiveness and safety; with further in vivo validation, they could become a key natural product material in developing therapeutics for sarcopenia.

## Figures and Tables

**Figure 1 pharmaceuticals-17-01661-f001:**
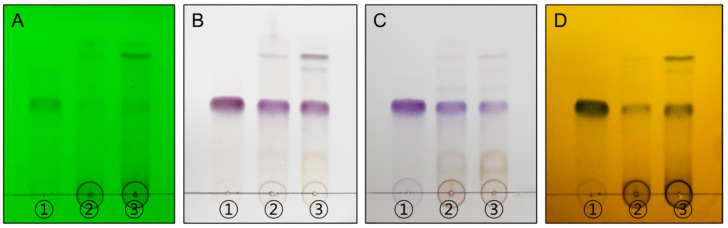
TLC chromatograms of each sample compound and the references: (**A**) UV 254 nm, (**B**) 10% H_2_SO_4_, (**C**) *ρ*-anisaldehyde H_2_SO_4_, and (**D**) FeCl_3_. The eluent system employed was chloroform/methanol/water = 70:30:4 (*v*/*v*/*v*). ① Oregonin, ② *Alnus japonica* ethanol extract (AJE), and ③ *Alnus japonica* hot water extract (AJHW).

**Figure 2 pharmaceuticals-17-01661-f002:**
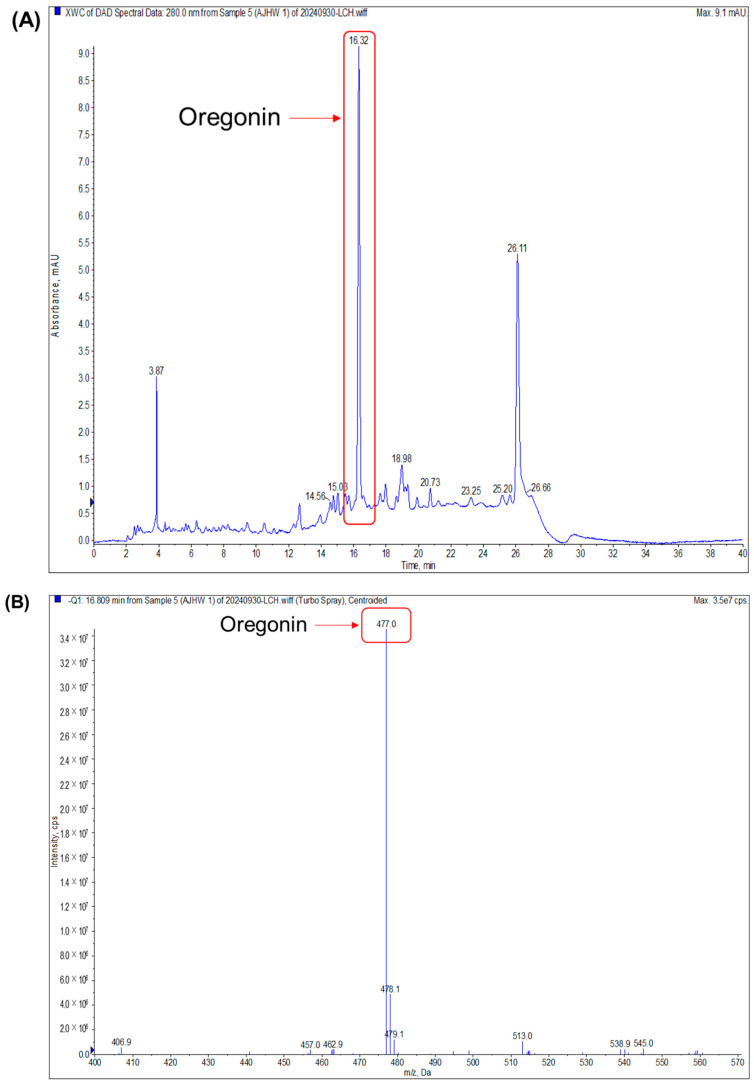
Negative mode LC-MS/MS analysis of oregonin (1000 μg/mL). (**A**) Extracted ion chromatogram of oregonin and (**B**) product ion mass spectrum of oregonin.

**Figure 3 pharmaceuticals-17-01661-f003:**
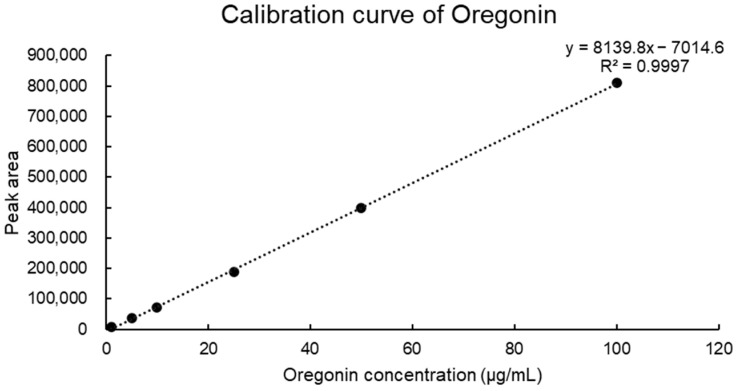
Calibration curve and equation of oregonin (100, 50, 25, 10, 5, and 1 μg/mL of oregonin).

**Figure 4 pharmaceuticals-17-01661-f004:**
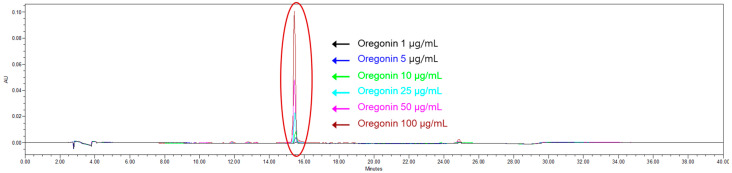
HPLC chromatogram of oregonin (100, 50, 25, 10, 5, and 1 μg/mL of oregonin).

**Figure 5 pharmaceuticals-17-01661-f005:**
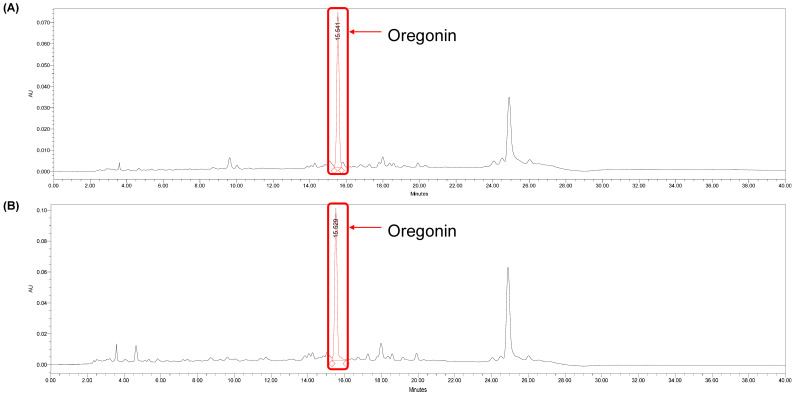
HPLC chromatogram of (**A**) *Alnus japonica* ethanol extract (AJE) (1000 μg/mL) and (**B**) *Alnus japonica* hot water extract (AJHW) (1000 μg/mL).

**Figure 6 pharmaceuticals-17-01661-f006:**
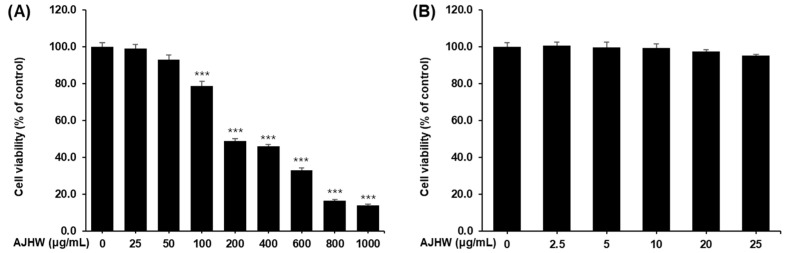
Cytotoxicity of *Alnus japonica* hot water extract (AJHW) on C2C12 myoblasts. (**A**) Treatment concentrations: 0, 25, 50, 100, 200, 400, 600, 800, and 1000 μg/mL. (**B**) Treatment concentrations: 0, 2.5, 5, 10, 20, and 25 μg/mL. Cell viability was calculated as described in Section 4. Values are expressed as the mean ± S.E.M. (*n* = 5). *** *p* < 0.001 significantly different from that of the 0 μg/mL group.

**Figure 7 pharmaceuticals-17-01661-f007:**
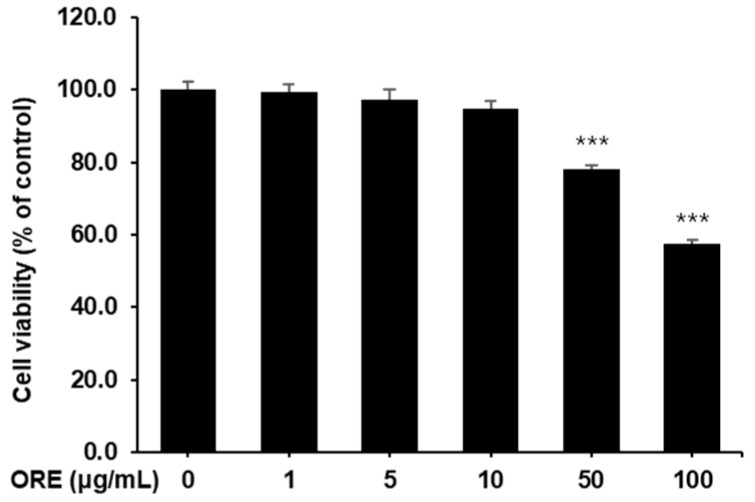
Cytotoxicity of oregonin (ORE) on C2C12 myoblasts. Cell viability was calculated as outlined in the Section 4. Values are expressed as the mean ± S.E.M. (*n* = 4). *** *p* < 0.001 indicating a significant difference from the 0 μg/mL group.

**Figure 8 pharmaceuticals-17-01661-f008:**
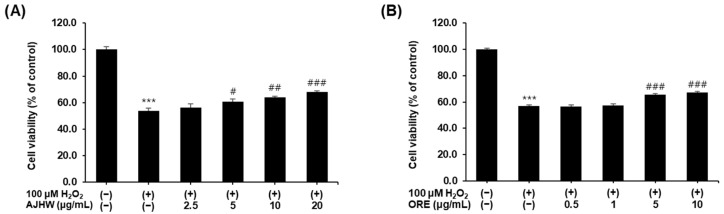
Protective effect of (**A**) *Alnus japonica* hot water extract (AJHW) and (**B**) oregonin (ORE) on cell viability in H_2_O_2_-treated C2C12 myoblasts. Values are expressed as the mean ± S.E.M. (*n* = 4). *** *p* < 0.001 significantly different from that of [H_2_O_2_ (−)/AJHW (−)] and [H_2_O_2_ (−)/ORE (−)] group. ^#^
*p* < 0.05, ^##^
*p* < 0.01, ^###^
*p* < 0.001 significantly different from that of the [H_2_O_2_ (+)/AJHW (−)] and [H_2_O_2_ (+)/ORE (−)] group.

**Figure 9 pharmaceuticals-17-01661-f009:**
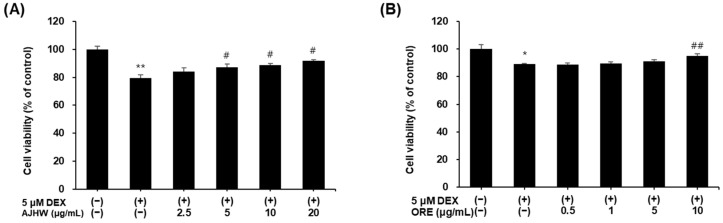
Protective effect of (**A**) *Alnus japonica* hot water extract (AJHW) and (**B**) oregonin (ORE) on cell viability in DEX-treated C2C12 myotubes. Values are expressed as the mean ± S.E.M. (*n* = 4). * *p* < 0.05, ** *p* < 0.01 significantly different from that of [DEX (−)/AJHW (−)], [DEX (−)/ORE (−)] group. ^#^ *p* < 0.05, ^##^ *p* < 0.01 significantly different from that of [DEX (+)/AJHW (−)], [DEX (+)/ORE (−)] group.

**Figure 10 pharmaceuticals-17-01661-f010:**
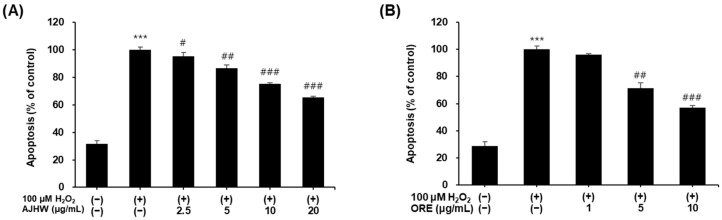
Protective effect of (**A**) *Alnus japonica* hot water extract (AJHW) and (**B**) oregonin (ORE) on apoptosis in H_2_O_2_-treated C2C12 myoblasts. Values are expressed as the mean ± S.E.M. (*n* = 4). *** *p* < 0.001 significantly different from that of [H_2_O_2_ (−)/AJHW (−)], [H_2_O_2_ (−)/ORE (−)] group. ^#^
*p* < 0.05, ^##^
*p* < 0.01, ^###^
*p* < 0.001 significantly different from those of the [H_2_O_2_ (+)/AJHW (−)] and [H_2_O_2_ (+)/ORE (−)] groups.

**Figure 11 pharmaceuticals-17-01661-f011:**
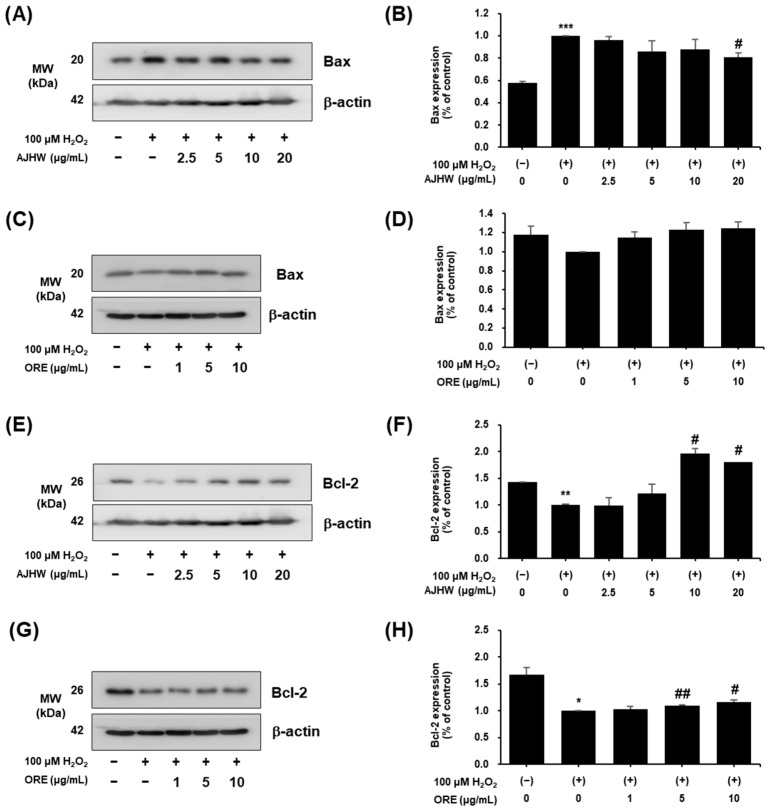
Anti-apoptotic effects of *Alnus japonica* hot water extract (AJHW) and oregonin (ORE) on H_2_O_2_-induced oxidative damage in C2C12 myoblasts. Western blotting was used to analyze the levels of (**A**–**D**) Bax, (**E**–**H**) Bcl-2, and β-actin, as described in Section 4. Values are expressed as the mean ± S.E.M. (*n* = 3). * *p* < 0.05, ** *p* < 0.01, *** *p* < 0.001 significantly different from that of [H_2_O_2_ (−)/AJHW (−)], [H_2_O_2_ (−)/ORE (−)] group. ^#^
*p* < 0.05, ^##^
*p* < 0.01 significantly different from those of the [H_2_O_2_ (+)/AJHW (-)] and [H_2_O_2_ (+)/ORE (−)] groups.

**Figure 12 pharmaceuticals-17-01661-f012:**
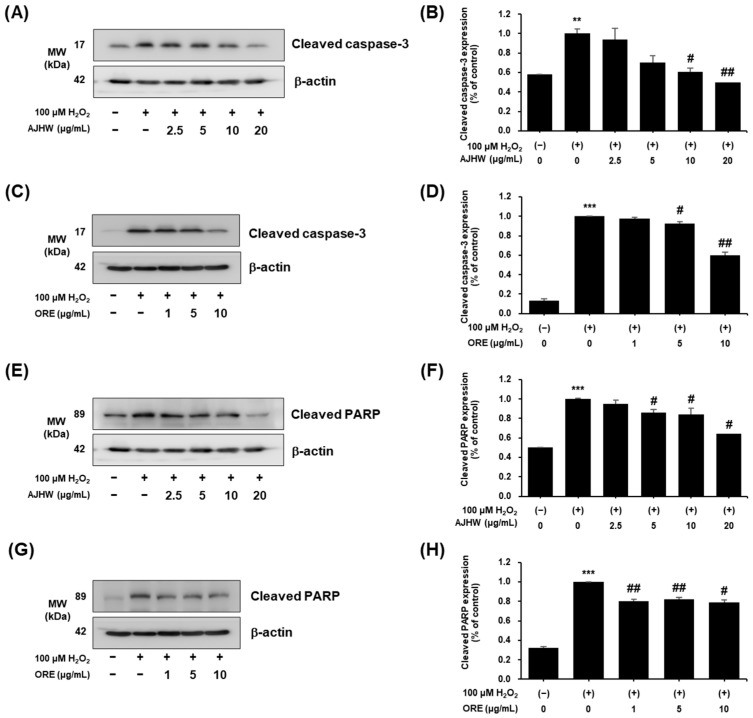
Anti-apoptotic effects of *Alnus japonica* hot water extract (AJHW) and oregonin (ORE) on H_2_O_2_-induced oxidative damage in C2C12. Western blotting was used to analyze the levels of (**A**–**D**) cleaved caspase-3, (**E**–**H**) cleaved PARP, and β-actin, as outlined in Section 4. Values are expressed as the mean ± S.E.M. (*n* = 3). ** *p* < 0.01, *** *p* < 0.001 significantly different from [H_2_O_2_ (−)/AJHW (−)], [H_2_O_2_ (−)/ORE (−)] group. ^#^
*p* < 0.05, ^##^
*p* < 0.01 significantly different from [H_2_O_2_ (+)/AJHW (−)] and [H_2_O_2_ (+)/ORE (−)] groups.

**Figure 13 pharmaceuticals-17-01661-f013:**
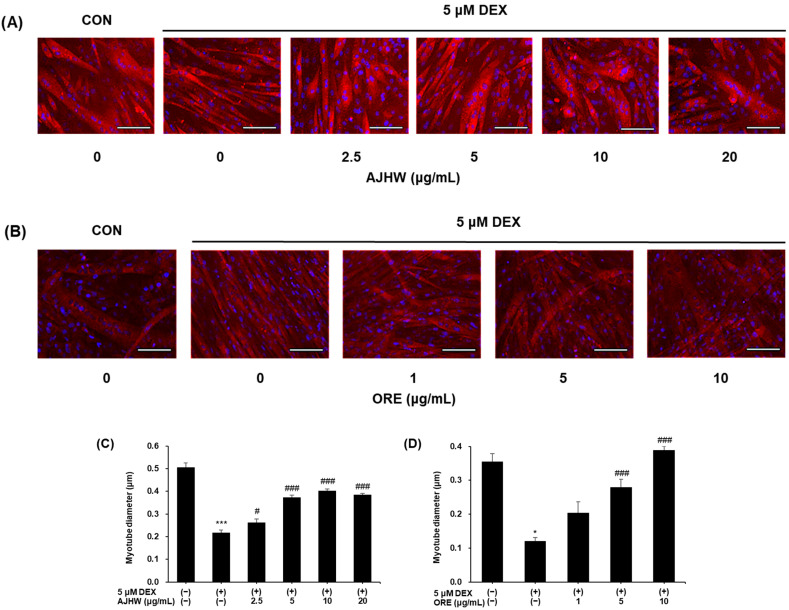
Effects of (**A**,**C**) *Alnus japonica* hot water extract (AJHW) and (**B**,**D**) oregonin (ORE) on dexamethasone-induced muscle atrophy in C2C12 myotubes. Values are expressed as the mean ± S.E.M. (*n* = 5). The scale bar represents 100 μm. * *p* < 0.05, *** *p* < 0.001 significantly different from that of [DEX (−)/AJHW (−)], [DEX (−)/ORE (−)] group. ^#^
*p* < 0.05, ^###^
*p* < 0.001 significantly different from that of [DEX (+)/AJHW (−)], [DEX (+)/ORE (−)] group.

**Figure 14 pharmaceuticals-17-01661-f014:**
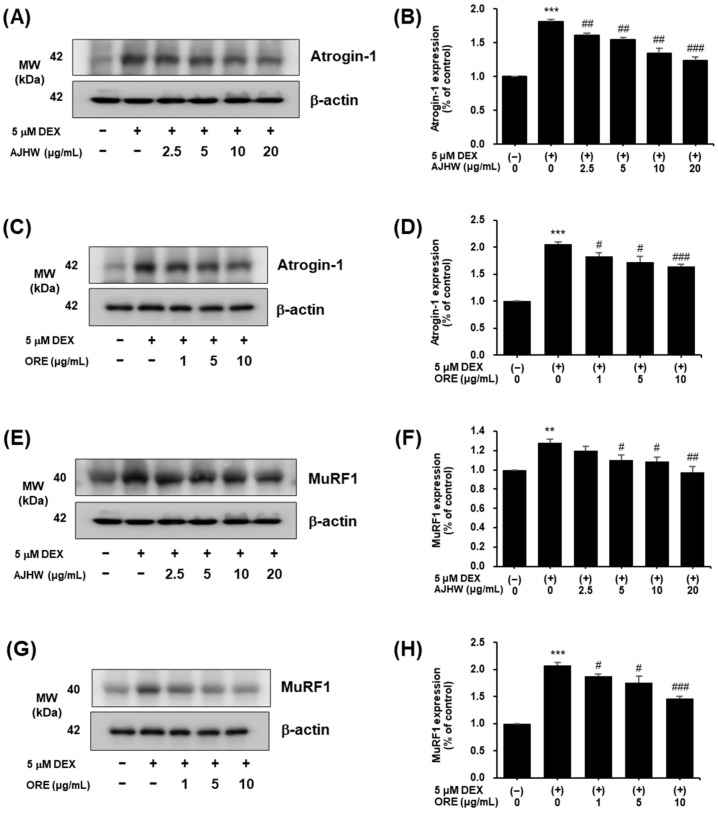
Effect of *Alnus japonica* hot water extract (AJHW) and oregonin (ORE) on the muscle-degradation-related protein expression levels of (**A**–**D**) Atrogin-1, (**E**–**H**) MuRF1 in DEX-treated C2C12 myotubes. AJHW and ORE were added to DEX-treated C2C12 myotubes and cultured for 24 h. Protein expression levels were determined using Western blotting. The expression levels were normalized to that of β-actin and expressed relative to those in the CON group. Values are expressed as the mean ± S.E.M. (*n* = 4). ** *p* < 0.01, *** *p* < 0.001 significantly different from [DEX (−)/AJHW (−)], [DEX (−)/ORE (−)] group. ^#^
*p* < 0.05, ^##^
*p* < 0.01, ^###^
*p* < 0.001 significantly different from the [DEX (+)/AJHW (−)], [DEX (+)/ORE (−)] group.

**Figure 15 pharmaceuticals-17-01661-f015:**
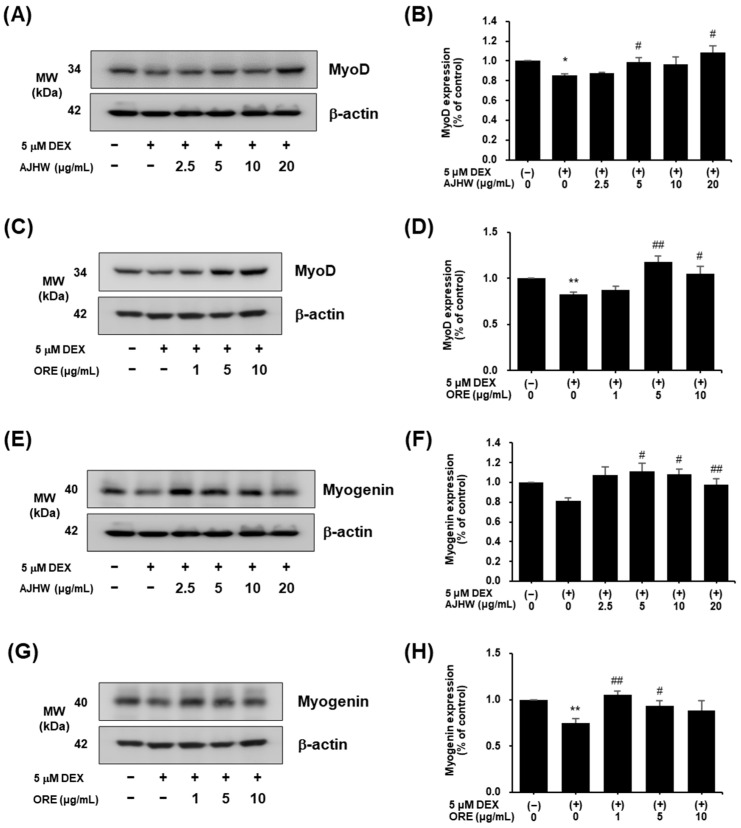
Effect of *Alnus japonica* hot water extract (AJHW) and oregonin (ORE) on the muscle-synthesis-related protein expression levels of (**A**–**D**) MyoD, and (**E**–**H**) Myogenin in DEX-treated C2C12 myotubes. AJHW and ORE were added to DEX-treated C2C12 myotubes and cultured for 24 h. Protein expression levels were determined using Western blotting. The protein expression levels were normalized to β-actin and expressed relative to those in the CON group. Values are expressed as the mean ± S.E.M. (*n* = 4). * *p* < 0.05, ** *p* < 0.01 significantly different from that of [DEX (−)/AJHW (−)], [DEX (−)/ORE (−)] group. ^#^
*p* < 0.05, ^##^
*p* < 0.01 significantly different from the [DEX (+)/AJHW (−)], [DEX (+)/ORE (−)] group.

**Figure 16 pharmaceuticals-17-01661-f016:**
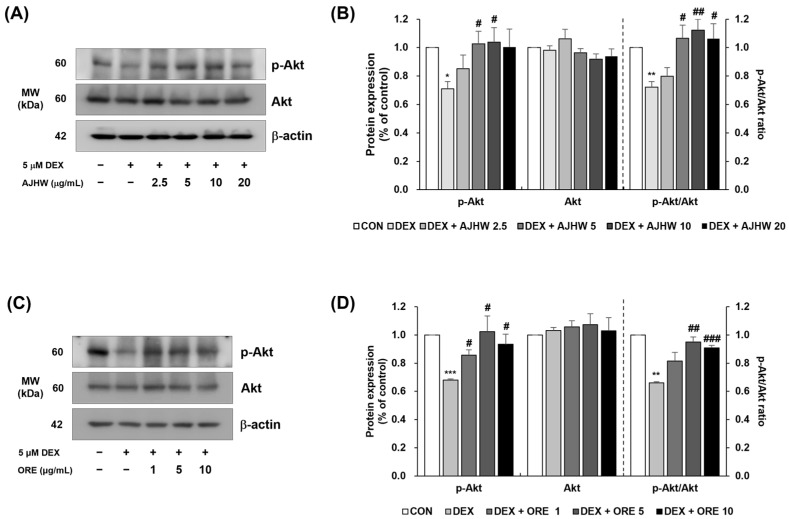
Effect of *Alnus japonica* hot water extract (AJHW) and oregonin (ORE) on the protein expression levels of (**A**–**D**) phospho-Akt and Akt, and (**E**–**H**) phospho-mTOR and mTOR in DEX-treated C2C12 myotubes. AJHW and ORE were separately added to these cells and incubated for 24 h. Protein expression was assessed via Western blotting, with β-actin as the normalization standard, and expressed relative to the CON group. Values are reported as the mean ± S.E.M. (*n* = 3). * *p* < 0.05, ** *p* < 0.01, *** *p* < 0.001, indicating significant differences from the [DEX (−)/AJHW (−)], [DEX (−)/ORE (−)] group. ^#^
*p* < 0.05, ^##^
*p* < 0.01, ^###^
*p* < 0.001 indicating significant differences from the [DEX (+)/AJHW (−)], [DEX (+)/ORE (−)] group.

**Figure 17 pharmaceuticals-17-01661-f017:**
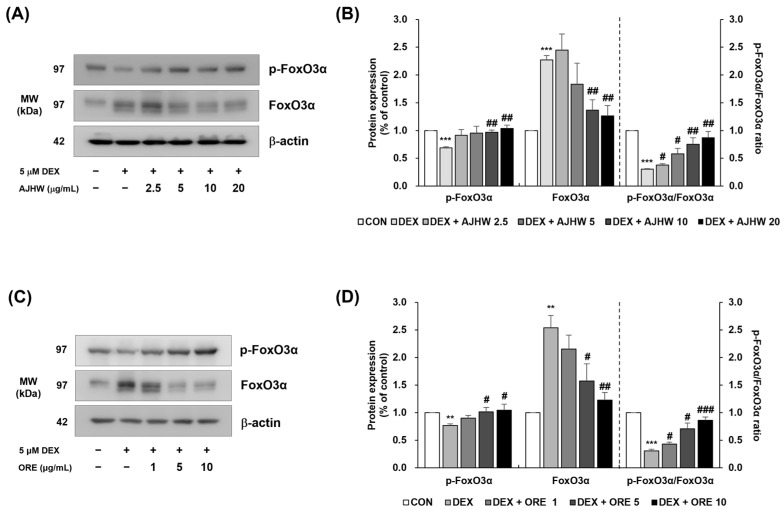
Effect of *Alnus japonica* hot water extract (AJHW) and oregonin (ORE) on the protein expression levels of (**A**–**D**) phospho-FoxO3α and FoxO3α in DEX-treated C2C12 myotubes. AJHW and ORE were separately added to these cells and incubated for 24 h. Protein expression was measured using Western blotting, normalized to β-actin, and reported relative to the CON group. Values are expressed as the mean ± S.E.M. (*n* = 3). ** *p* < 0.01, *** *p* < 0.001 indicating significant differences from the [DEX (−)/AJHW (−)], [DEX (−)/ORE (−)] group. ^#^
*p* < 0.05, ^##^
*p* < 0.01, ^###^
*p* < 0.001 indicating significant differences from the [DEX (+)/AJHW (−)], [DEX (+)/ORE (−)] group.

**Figure 18 pharmaceuticals-17-01661-f018:**
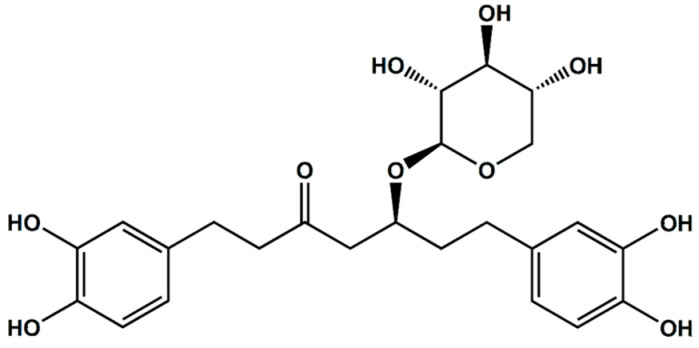
The structure of oregonin. The chemical structure was illustrated using ChemDraw Ultra 7.0 (CambridgeSoft, Cambridge, MA, USA).

**Table 1 pharmaceuticals-17-01661-t001:** Effect of *Alnus japonica* hot water extract on muscle-degradation- and synthesis-related gene expression.

DEX (5 μM)	AJHW (μg/mL)	mRNA			
		*Atrogin-1*	*MuRF1*	*MyoD*	*Myogenin*
−	−	0.10 ± 0.02	0.10 ± 0.02	3.10 ± 0.19	2.19 ± 0.23
+	−	1.00 ± 0.09 ***	1.00 ± 0.11 ***	1.00 ± 0.29 ***	1.00 ± 0.07 ***
+	2.5	1.02 ± 0.17	0.93 ± 0.15	1.38 ± 0.12	1.28 ± 0.17
+	5	0.74 ± 0.13	0.67 ± 0.12	1.55 ± 0.18	1.88 ± 0.22 ^##^
+	10	0.67 ± 0.04 ^#^	0.61 ± 0.03 ^##^	1.60 ± 0.09	2.19 ± 0.21 ^###^
+	20	0.51 ± 0.03 ^##^	0.46 ± 0.03 ^###^	1.98 ± 0.15 ^#^	1.69 ± 0.09 ^###^

DEX: dexamethasone; AJHW: *Alnus japonica* hot water extract. Values are presented as the mean ± S.E.M. (*n* = 6). mRNA expression of the target was normalized to that of GAPDH. *** *p* < 0.001, indicating significant difference from the [DEX (−)/AJHW (−)] group. ^#^
*p* < 0.05, ^##^
*p* < 0.01, ^###^
*p* < 0.001, significantly different from the [DEX (+)/AJHW (−)] group.

**Table 2 pharmaceuticals-17-01661-t002:** Effect of oregonin on muscle-degradation- and synthesis-related gene expression.

DEX (5 μM)	ORE (μg/mL)	mRNA			
		*Atrogin-1*	*MuRF1*	*MyoD*	*Myogenin*
−	−	0.09 ± 0.02	0.11 ± 0.02	3.32 ± 0.33	3.09 ± 0.37
+	−	1.00 ± 0.09 ***	1.00 ± 0.12 ***	1.00 ± 0.16 ***	1.00 ± 0.10 **
+	1	0.67 ± 0.10 ^#^	0.92 ± 0.11	1.26 ± 0.14	1.36 ± 0.19 ^#^
+	5	0.57 ± 0.08 ^##^	0.75 ± 0.05	2.06 ± 0.23 ^##^	2.03 ± 0.22 ^##^
+	10	0.38 ± 0.06 ^###^	0.36 ± 0.05 ^###^	2.72 ± 0.23 ^###^	2.29 ± 0.15 ^###^

DEX: dexamethasone; ORE: Oregonin. Values are presented as the mean ± S.E.M. (*n* = 6). mRNA expression of the target was normalized to that of GAPDH. ** *p* < 0.01, *** *p* < 0.001, significantly different from the [DEX (−)/ORE (−)] group. ^#^
*p* < 0.05, ^##^
*p* < 0.01, ^###^
*p* < 0.001, significantly different from the [DEX (+)/ORE (−)] group.

**Table 3 pharmaceuticals-17-01661-t003:** Specific primer sequences for RT-PCR.

mRNA	Primer Sequence		GeneBank No.
*Atrogin-1*	Forward	5′-GCCCTCCACACTAGTTGACC-3′	NM_026346.3
	Reverse	5′-GACGGATTGACAGCCAGGAA-3′	
*MuR* *F* *1*	Forward	5′-GAGGGCCATTGACTTTGGGA-3′	NM_001039048.2
	Reverse	5′-TTTACCCTCTGTGGTCACGC-3′	
*MyoD1*	Forward	5′-GCACTACAGTGGCGACTCAGAT-3′	NM_010866.2
	Reverse	5′-TAGTAGGCGGTGTCGTAGCCAT-3′	
*Myogenin*	Forward	5′-CCATCCAGTACATTGAGCGCCT-3′	NM_031189.2
	Reverse	5′-CTGTGGGAGTTGCATTCACTGG-3′	
*GAPDH*	Forward	5′-TGGGTGTGAACCATGAGAAG-3′	NM_008084.3
	Reverse	5′-GCTAAGCAGTTGGTGGTGC-3‘	

## Data Availability

The original contributions presented in this study are included in the article; further inquiries can be directed to the corresponding author.

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
