# Peer review of "Effects of Alnus japonica Hot Water Extract and Oregonin on Muscle Loss and Muscle Atrophy in C2C12 Murine Skeletal Muscle Cells"

_pharmaceuticals, 2024, doi:10.3390/ph17121661_

Round 1
Reviewer 1 Report
Comments and Suggestions for Authors
The authors have presented in vitro data using C2C12 murine muscle cells to demonstrate the therapeutics effects of AJHW and its active ingredient oregonin on various pathological features observed in sarcopenia such as apoptosis, muscle viability, muscle protein degradation and muscle protein synthesis. They present a detailed look at molecular signatures for several pathways implicated in sarcopenia. However, at this time this is an in vitro study describing a new therapeutic strategy for sarcopenia and proof of concept in vivo studies are pending.
Minor
1. Authors have established in vitro cytotoxic concentrations for AJHW and Oregonin however, the translatability of this therapeutic strategy to animal models of sarcopenia and ultimately to human patients is not discussed. Can the authors shed light on whether AJHW is safe for treatment or would likely need further modifications.
2. Figures 13A and 13B are blurry. Authors should upload better quality images as differences in myotube diameters are not very apparent.
3. Line 590, do authors mean to say "....suggesting that Oregonin does not influence Bax expression in these cells." It seems like H2O2 is incorrectly placed here as a typo. Authors should also explain why they think Oregonin does not have an effect on Bax.
Author Response
Thank you very much for taking the time to review this manuscript. Please find the detailed responses below and the corresponding revisions/corrections highlighted/in track changes in the re-submitted files
Comments 1: Authors have established in vitro cytotoxic concentrations for AJHW and Oregonin however, the translatability of this therapeutic strategy to animal models of sarcopenia and ultimately to human patients is not discussed. Can the authors shed light on whether AJHW is safe for treatment or would likely need further modifications.
Response 1: Thank you for your valuable feedback. Our research team completed preclinical studies on Ulmus macrocarpa extract at both in vivo and in vitro stages in June and August of this year [1, 2], and human application trials for this extract are scheduled to begin in January. We agree that discussing the translatability of our findings is important. The reason we chose hot water extraction for the extraction of Alnus japonica is that we ultimately aim for the commercialization of pharmaceuticals. Accordingly, in addition to the in vitro studies presented in this manuscript, we have conducted in vivo research on the safety and efficacy of AJHW and Oregonin in animal models, and the results are currently under review in a separate manuscript. We believe these complementary studies will provide further insights into the therapeutic potential and safety of AJHW for sarcopenia treatment. Thank you for highlighting this aspect.
- Lee CH, Kwon Y, Park S, Kim T, Kim MS, Kim EJ, Jung JI, Min S, Park KH, Jeong JH, Choi SE. The Impact of Ulmus macrocarpa Extracts on a Model of Sarcopenia-Induced C57BL/6 Mice. Int J Mol Sci. 2024 Jun 4;25(11):6197. doi: 10.3390/ijms25116197. PMID: 38892385; PMCID: PMC11172872.
- Kim MS, Park S, Kwon Y, Kim T, Lee CH, Jang H, Kim EJ, Jung JI, Min S, Park KH, Choi SE. Effects of Ulmus macrocarpa Extract and Catechin 7-O-β-D-apiofuranoside on Muscle Loss and Muscle Atrophy in C2C12 Murine Skeletal Muscle Cells. Curr Issues Mol Biol. 2024 Aug 1;46(8):8320-8339. doi: 10.3390/cimb46080491. PMID: 39194708; PMCID: PMC11352752.
Comments 2: Figures 13A and 13B are blurry. Authors should upload better quality images as differences in myotube diameters are not very apparent.
Response 2: Thank you for pointing out the issue with Figures 13A and 13B. We have replaced them with higher-resolution images to ensure the differences in myotube diameters are clearly visible. The updated figures are included in the revised manuscript. We appreciate your feedback to improve the clarity of our work.
Comments 3: Line 590, do authors mean to say "....suggesting that Oregonin does not influence Bax expression in these cells." It seems like H2O2 is incorrectly placed here as a typo. Authors should also explain why they think Oregonin does not have an effect on Bax.
Response 3: We appreciate the reviewer’s careful observation. You are correct that the placement of Hâ‚‚Oâ‚‚ in line 590 was a typo. The sentence has been revised to accurately state, “...suggests that oregonin does not influence Bax expression in these cells.” and moved to line 600-601 with the added content.
Thank you for your thoughtful and valid comment. Upon re-evaluating the manuscript, we fully agree that the reasoning for why Oregonin does not have an effect on Bax should have been included, and we appreciate you pointing out this oversight. In our study, AJHW demonstrated significant effects on all apoptosis-related biomarkers. Based on these findings, we tested Oregonin, a key active compound and marker of the Alnus genus, and observed a clear correlation with most apoptosis-related biomarkers. However, Bax expression did not show a significant response to Oregonin treatment, leading us to conclude that Oregonin alone does not influence Bax expression.
Despite this, the strong correlations with other apoptosis-related biomarkers confirm the substantial anti-apoptotic activity of Oregonin. Additionally, we hypothesized that the significant effects observed in AJHW were likely due to the synergistic action of other minor compounds present in the extract other than Oregonin [3]. This review has been added to the Discussion Section (Line 598-604). To further investigate this, we also plan to explore the effects of additional diarylheptanoid compounds on Bax expression in future studies. Unfortunately, due to current constraints, this analysis cannot be conducted immediately, but it will be prioritized in subsequent research.
Thank you again for bringing this to our attention and for providing valuable feedback to improve the clarity and completeness of our work.
- Caesar LK, Cech NB. Synergy and antagonism in natural product extracts: when 1 + 1 does not equal 2. Nat Prod Rep. 2019 Jun 19, 36(6):869-888. doi: 10.1039/c9np00011a.

Reviewer 2 Report
Comments and Suggestions for Authors
As stated by the Authors, the protective effect of oregonin on sarcopenia has been already investigated in a work published in 2023. Therefore, this study does not seem particularly original. However, the Authors have proposed a novel extraction protocol that might be interesting and also detected that oregonin might represent a more efficient factor protecting muscle from sarcopenia as compared to other natural products already tested. In addition, they have investigated the effect of some novel biomarkers related to protein degradation in skeletal muscle that apparently were not tested in the work mentioned previously.
For these reasons, this manuscript might represent an interesting contribution. However, to improve the clarity and soundness of it I am suggesting a series of major mandatory revisions.
_Introduction is way too long and it seems to include a description (as opposed to a mere mention) of the results of the different extraction protocols (ethanol vs hot water). Accordingly, portions of this text should be moved to the Results section, or Discussion. That said, also the Discussion is pretty extended and this is detrimental to the overall clarity of the paper.
Line 116 “the study did not sufficiently explore muscle apoptosis-related biomarkers”. Please explain clearly which markers have been tested in that previous work (Ref. n.52) and state exactly which novel markers have been tested in the present work. Any significant differences emerging in comparison with Ref. 52?
_lines 218-219: “various concentrations of AJHW (0–1,000 μg/mL)…”…what is the estimated concentration of oregonin in these extracts? The Authors have identified a “safe” range of concentrations from the cell viability experiments with AJWH, i.e. up to 20 ug/ml. and it would be interesting to know how much oregonin they contain and compare it with the 10 ug/ml of pure oregonin they have also used.
_Another question related to the previous one; do the AJHW extracts used in this study likely contain also other potentially bioactive molecules? Maybe there is something obvious I missed, but either way it is important that you explain this point better. In other words, are the AJHW extracts a cocktail of molecules or, rather, they mostly contain oregonin?
_Also, is oregonin more efficiently enriched in the ethanol or hot-water extracts? From lines 140-142, hot-water extraction seems more efficient but please report how much exactly?
_In Fig.6 and in other parts of the ms, it must be always specified when “C2C12 cells” refer to myoblasts or differentiated myotubes.
_No images of myoblasts or myotubes are provided. I believe that at least some representative images for the various conditions analyzed should be reported. Possibly showing the beneficial effects of oregonin on said cells?
_Fig.11. I think that some of the markers analyzed should be tested also upon treatment with AJHW and pure oregonin solutions on C2C12 myoblasts and myotubes not subjected to H2O2-induced oxidative stress. The same with the markers tested for Muscle-Degradation and Synthesis-Related Gene Expression or for the Akt/mTOR/FoxO Signaling Pathway-Related. Basically, all the markers must be tested, at the various AJHW or oregonin concentrations, also on myotubes not treated with dexamethasone.
These controls might be important for possible pharmacological and therapeutical applications in the future.
_A not irrelevant weakness of this work is that no hints are given over the possible molecular mechanism of action. For example, is oregonin supposed to bind a protein/enzyme? Any ideas at least?
_Line 617: change “PCR” to RT-PCR.
Author Response
* Please see the attachment. (The revised Response to Comments 3 includes a Supplementary Figure for clarification. Please refer to the attached file for further details.)
Reviewer 2
Thank you very much for taking the time to review this manuscript. Please find the detailed responses below and the corresponding revisions/corrections highlighted/in track changes in the re-submitted files
Comments 1: Introduction is way too long and it seems to include a description (as opposed to a mere mention) of the results of the different extraction protocols (ethanol vs hot water). Accordingly, portions of this text should be moved to the Results section, or Discussion. That said, also the Discussion is pretty extended and this is detrimental to the overall clarity of the paper.
Response 1: Thank you for your feedback. The reason we mentioned the comparison of extraction protocols in the Introduction was to emphasize that different solvents yield distinct extracts, which is a crucial consideration in natural product research. However, we acknowledge that this information is already discussed in the Results section. In line with your suggestion, we have removed the detailed description from the Introduction to enhance the overall clarity and organization of the manuscript. Additionally, we wanted to highlight the advantages of reduced extraction time, so we have briefly added this point in the Discussion section (Line 545-548). Thank you for helping us improve our work.
Comments 2: Line 116 “the study did not sufficiently explore muscle apoptosis-related biomarkers”. Please explain clearly which markers have been tested in that previous work (Ref. n.52) and state exactly which novel markers have been tested in the present work. Any significant differences emerging in comparison with Ref. 52?
Response 2: Thank you for your insightful comment. The previous study (Ref. n.52) investigated various markers related to muscle synthesis and atrophy, including myosin heavy chain (MHC), MyoD, myogenin, myostatin, MuRF1, p38 MAPK, phospho-p38 MAPK, mTOR, phospho-mTOR, FoxO3a, phospho-FoxO3a, NF-κB, phospho-NF-κB, and MAFbX (Muscle Atrophy F-box, Atrogin-1). However, it did not thoroughly explore the expression of muscle synthesis-related signaling markers Akt and p-Akt, nor did it investigate apoptosis-related biomarkers. Notably, Akt and p-Akt are key components of the Akt/mTOR/FoxO signaling pathway, which plays a crucial role in muscle protein synthesis [1]. Additionally, muscle cell apoptosis contributes significantly to muscle atrophy and is one of the primary causes of sarcopenia [2]. This makes the investigation of apoptosis particularly important in the development and approval of therapeutics for sarcopenia.
In contrast to the previous study, we examined the expression of Akt and p-Akt, as well as apoptosis-related biomarkers, including Bax, Bcl-2, Cleaved-PARP, and Cleaved-Caspase3. Our research provides new insights into these biomarkers and enhances understanding of the therapeutic potential of AJHW and oregonin in addressing sarcopenia.
- Bonifacio, A.; Sanvee, G.M.; Bouitbir, J.; Krähenbühl, S. The AKT/mTOR signaling pathway plays a key role in statin-induced myotoxicity. Biochimica et Biophysica Acta (BBA) - Molecular Cell Research 2015, 1853, 1841-1849, doi:https://doi.org/10.1016/j.bbamcr.2015.04.010.
- Alway, S.E.; Siu, P.M. Nuclear apoptosis contributes to sarcopenia. Exerc Sport Sci Rev 2008, 36, 51-57, doi:10.1097/JES.0b013e318168e9dc
Comments 3: _lines 218-219: “various concentrations of AJHW (0–1,000 μg/mL)…”…what is the estimated concentration of oregonin in these extracts? The Authors have identified a “safe” range of concentrations from the cell viability experiments with AJWH, i.e. up to 20 ug/ml. and it would be interesting to know how much oregonin they contain and compare it with the 10 ug/ml of pure oregonin they have also used.
Response 3:
Thank you for your interest in such detailed aspects of our study. In this study, HPLC analysis confirmed that AJHW at a concentration of 1,000 μg/mL contains 53.52 ± 0.21 μg/mL (n = 3) of oregonin. Typically, direct comparisons between single compounds and extracts containing those compounds at equivalent ppm concentrations are not standard practice. However, in response to your thoughtful query, we explored this further. From six concentrations of oregonin, a standard calibration curve and curve equation were derived, yielding Y = 8139.8x – 7014.6 (R2 = 0.9997) increments of oregonin (S. 1A). As anticipated, the high-purity 10 μg/mL pure oregonin was measured at 9.59 ppm (S. 1B), while AJHW at 20 μg/mL contained oregonin at a concentration of 3.01 ± 0.14 ppm (n = 3), constituting approximately 15% of the extract (S. 1C).
Supplementary Figure 1. (A) Calibration curve and equation of oregonin (100, 50, 25, 10, 5, and 1 μg/mL of oregonin, y = 8139.8x - 7014.6 (R² = 0.9997)), (B) HPLC chromatogram of oregonin (100, 50, 25, 10, 5, and 1 μg/mL of oregonin), (C) HPLC chromatogram of AJHW 20 μg/mL.
Comments 4: _Another question related to the previous one; do the AJHW extracts used in this study likely contain also other potentially bioactive molecules? Maybe there is something obvious I missed, but either way it is important that you explain this point better. In other words, are the AJHW extracts a cocktail of molecules or, rather, they mostly contain oregonin?
Response 4: Thank you for your thoughtful question and for pointing out an area that could benefit from further clarification. Oregonin has been identified as the major compound in Alnus species through numerous previous studies. Furthermore, chemotaxonomic research has confirmed oregonin as a primary component and an indicator compound in these plants [3]. However, oregonin is not the only compound present, as other diarylheptanoid compounds have also been reported to exist in trace amounts in Alnus species [4,5]. I realized that I may not have explained this point in sufficient detail earlier, and I appreciate your highlighting it. In response, our research team conducted HPLC analysis of AJHW, which revealed that oregonin is the most abundant compound in the extract. Therefore, while AJHW is a mixture, it can be considered an extract primarily composed of oregonin. I have added this clarification to Conclusion section (Line 899-900).
- Choi, S.E. Chemotaxonomic Significance of Oregonin in Alnus Species. Asian Journal of Chemistry 2013, 25, 6989-6990, doi:10.14233/ajchem.2013.15090
- Choi, S.E., Park, K.H., Kim, M.H., Song, J.H., Jin, H.Y., Lee, M.W. Diarylheptanoids from the Bark of Alnus pendula Matsumura. Natural Product Sciences, 2012, 18(2), 106-110. doi: http://dx.doi.org/
- Vidaković, V.; Novaković, M.; Popović, Z.; Janković, M.; Matić, R.; Tešević, V.; Bojović, S. Significance of diarylheptanoids for chemotaxonomical distinguishing between Alnus glutinosa and Alnus incana. Holzforschung 2018, 72, 9-16, doi:doi:10.1515/hf-2017-0074
Comments 5: _Also, is oregonin more efficiently enriched in the ethanol or hot-water extracts? From lines 140-142, hot-water extraction seems more efficient but please report how much exactly?
Response 5: Based on our results, oregonin content was 38.6% higher in the hot-water extracts compared to the ethanol extracts, as confirmed by HPLC analysis. This significant increase highlights the superior efficiency of hot-water extraction. We appreciate your suggestion and have included these specific details in the revised manuscript (Line 543-545).
Comments 6: _In Fig.6 and in other parts of the ms, it must be always specified when “C2C12 cells” refer to myoblasts or differentiated myotubes.
Response 6: Thank you for pointing out the part I missed. We have reviewed and revised the manuscript to ensure that "C2C12 cells" are clearly specified as either myoblasts or myotubes in all relevant sections.
Comments 7: _No images of myoblasts or myotubes are provided. I believe that at least some representative images for the various conditions analyzed should be reported. Possibly showing the beneficial effects of oregonin on said cells?
Response 7: Thank you for your comment. Unfortunately, we currently only have images used to measure the diameter of C2C12 myotubes. To address your point, we have referenced a representative image from a related study that illustrates the effects of Dexamethasone and subsequent treatment on myotube morphology [6]. In future studies, we will ensure to prepare and include a comprehensive set of images showing various conditions, as suggested.
- Jiang R, Wang M, Shi L, Zhou J, Ma R, Feng K, Chen X, Xu X, Li X, Li T, Sun L. Panax ginseng Total Protein Facilitates Recovery from Dexamethasone-Induced Muscle Atrophy through the Activation of Glucose Consumption in C2C12 Myotubes. Biomed Res Int. 2019 Aug 6; 2019:3719643. doi: 10.1155/2019/3719643.
Comments 8: _Fig.11. I think that some of the markers analyzed should be tested also upon treatment with AJHW and pure oregonin solutions on C2C12 myoblasts and myotubes not subjected to H2O2-induced oxidative stress. The same with the markers tested for Muscle-Degradation and Synthesis-Related Gene Expression or for the Akt/mTOR/FoxO Signaling Pathway-Related. Basically, all the markers must be tested, at the various AJHW or oregonin concentrations, also on myotubes not treated with dexamethasone.
These controls might be important for possible pharmacological and therapeutical applications in the future.
Response 8: We greatly appreciate this suggestion. In this study, we primarily focused on assessing the efficacy of AJHW and oregonin under pathological conditions— H2O2-induced oxidative stress to model apoptosis and Dexamethasone-induced aging to mimic muscle atrophy. These approaches allowed us to evaluate the therapeutic potential of AJHW and oregonin in diseased states.
While we understand the importance of testing AJHW and oregonin under normal physiological conditions to further explore their pharmacological applications, these experiments were not included in the current study due to its design and scope. However, we recognize the value of this perspective and will consider it in future studies. Thank you for bringing this important consideration to our attention.
Comments 9: _A not irrelevant weakness of this work is that no hints are given over the possible molecular mechanism of action. For example, is oregonin supposed to bind a protein/enzyme? Any ideas at least?
Response 9: Thank you for your insightful comment. We fully agree with your observation, and we appreciate you bringing attention to this aspect that we had not addressed in detail.
AJHW and oregonin are potentially involved in pathways such as Akt/mTOR/FoxO signaling. Cross-validation through RT-PCR and Western blotting assays in our study demonstrated that oregonin plays a crucial role in suppressing muscle protein degradation markers, such as Atrogin-1 and MuRF1, by modulating this signaling pathway. This suggests that oregonin exerts its effects by influencing key intracellular signaling cascades, which are pivotal in muscle atrophy regulation. Furthermore, this study confirms that oregonin influences the cell signaling pathways in a bidirectional manner, acting to upregulate positive signals and downregulate negative signals. This review has been added to the Discussion Section (Line 670-675). Thank you for your valuable feedback.
Comments 10: _Line 617: change “PCR” to RT-PCR.
Response 10: Thank you for pointing out the oversight. The term has been corrected to "RT-PCR" as suggested.

Round 2
Reviewer 2 Report
Comments and Suggestions for Authors
I reckon that the Authors have sufficiently improved their manuscript based on some of my comments. I am afraid that the myotubes figure doesn’t look particularly improved (it might be in the PDF?). However, the myotubes still look blurry, as suggested by Reviewer 1. These images must be improved further.
I suggest to add a sort of disclaimer sentence to the Discussion or Conclusions. Something like “Further work is warranted to ascertain the exact molecular mechanism of oregonin, identifying which enzyme(s) or transcription factor(s) in the cytosol or nucleus are exactly bound or modified by oregonin.”
Author Response
Comments 1: I reckon that the Authors have sufficiently improved their manuscript based on some of my comments. I am afraid that the myotubes figure doesn‘t look particularly improved (it might be in the PDF?). However, the myotubes still look blurry, as suggested by Reviewer 1. These images must be improved further.
I suggest to add a sort of disclaimer sentence to the Discussion or Conclusions. Something like “Further work is warranted to ascertain the exact molecular mechanism of oregonin, identifying which enzyme(s) or transcription factor(s) in the cytosol or nucleus are exactly bound or modified by oregonin.”
Response 1: We fully acknowledge your concerns regarding the resolution of the myotube images and have made every effort to improve them. However, due to technical limitations inherent in the imaging process, further enhancement is challenging. As such, we have selected representative images, as done in prior relevant studies, and edited them while striving to enhance their resolution. We kindly ask for your understanding on this matter.
Additionally, as per your suggestion, we have included the following statement in the Discussion section (Line 675-682):
" However, this study has limitations in that it was unable to clearly elucidate the molecular mechanisms related to the intracellular effects of oregonin. Therefore, further studies are warranted to elucidate the precise molecular mechanisms of oregonin, particularly by identifying cytosolic or nuclear enzymes, or transcription factors specifically bound to or modified by oregonin. Furthermore, it has been confirmed that other diarylheptanoid compounds, in addition to oregonin, are present in Alnus japonica hot water extract, and further research into the mechanisms of action of these compounds is also needed.”
We appreciate your insightful suggestion, as it pointed out an important aspect we had overlooked. We believe the proposed statement aligns well with the scope of the manuscript while effectively emphasizing the necessity for future research.
Thank you sincerely for your valuable insights and constructive suggestions.
